# LAYERWISE RECURRENT ROUTER FOR MIXTURE-OF-EXPERTS

[1]**Zihan Qiu**[*]   [2]**Zeyu Huang**[*]   [3]**Shuang Cheng**   [4]**Yizhi Zhou**   [5]**Zili Wang**
[2,6]**Ivan Titov**   [7]**Jie Fu**[†]
[1]Alibaba Group,   [2]University of Edinburgh,   [3]ICT, Chinese Academy of Sciences,
[4]Nanjing University,   [5]INF Technology   [6]University of Amsterdam   [7]Shanghai AI Lab
qzh11628@gmail.com, zeyu.huang@ed.ac.uk, fujie@pjlab.org.cn

## ABSTRACT

The scaling of large language models (LLMs) has revolutionized their capabilities in various tasks, yet this growth must be matched with efficient computational strategies. The Mixture-of-Experts (MoE) architecture stands out for its ability to scale model size without significantly increasing training costs. Despite their advantages, current MoE models often display parameter inefficiency. For instance, a pre-trained MoE-based LLM with 52 billion parameters might perform comparably to a standard model with 6.7 billion parameters (Rajbhandari et al., 2022). Being a crucial part of MoE, current routers in different layers independently assign tokens without leveraging historical routing information, potentially leading to suboptimal token-expert combinations and the parameter inefficiency problem. To alleviate this issue, we introduce the Layerwise Recurrent Router for Mixture-of-Experts (RMoE). RMoE leverages a Gated Recurrent Unit (GRU) to establish dependencies between routing decisions across consecutive layers. Such layerwise recurrence can be efficiently parallelly computed for input tokens and introduces negotiable costs. Our extensive empirical evaluations demonstrate that RMoE-based language models consistently outperform a spectrum of baseline models. Furthermore, RMoE integrates a novel computation stage orthogonal to existing methods, allowing seamless compatibility with other MoE architectures. Our analyses attribute RMoE's gains to its effective cross-layer information sharing, which also improves expert selection and diversity. Our code is at https://github.com/qiuzh20/RMoE.

## 1   INTRODUCTION

In the era of large language models (LLMs), scaling the model parameters and training data up has unlocked remarkable model capabilities, such as in-context learning (Brown et al., 2020; Dong et al., 2022), nuanced conversations (Ouyang et al., 2022), and even complex code (Guo et al., 2024) and math (Imani et al., 2023) tasks. These advancements showcase the profound impact of increasing model size. The quest to enhance neural networks' capacity while ensuring training and inference efficiency spurred the development of computation-efficient transformer architectures. The Mixture-of-Experts (MoE) framework is one of such efficient architectural recipes (Shazeer et al., 2017; Lepikhin et al., 2021; Fedus et al., 2022; Zhang et al., 2022; Dai et al., 2024). Most MoE modules comprise one *router* and a group of *expert* networks. The router, usually parametrized as one linear layer, conditionally and sparsely assigns each input token to its corresponding experts, *i.e.*, the FeedForward Network (FFN) in the transformer layer. Therefore, MoE can significantly scale the model size and keep computational costs nearly unchanged (Smith et al., 2022).

Despite efficiently increasing the model size, most current pre-trained MoE models are not on par with standard models of the same size, demonstrating their parameter inefficiency. For example, Rajbhandari et al. (2022) shows that with the same training data, an MoE with 52B parameters and 1.3B activated ones for each token performs similarly to a 6.7B standard model. Komatsuzaki

---

[*] Equal contribution
[†] Corresponding author

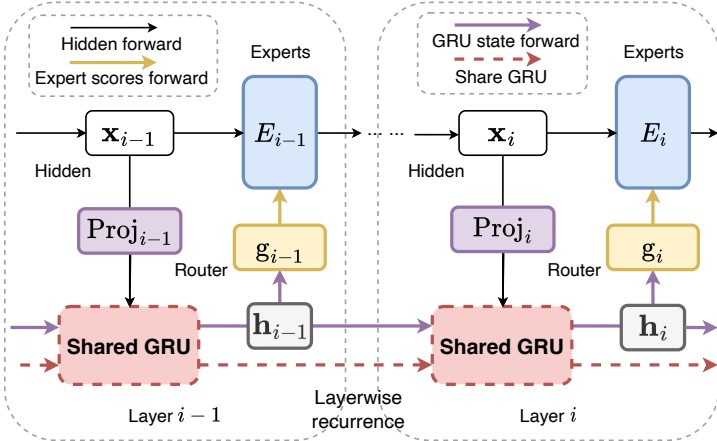

Figure 1: Recurrent router for Mixture-of-Experts. In the $i$-th layer, the hidden state $\mathbf{x}_i$ is **I.** projected to $\mathbf{x}'$ with alower hidden dimension (Eq. 4), **II.** combined with previous layer's GRU output $\mathbf{h}_{i-1}$, and processed through the cross-layer-shared GRU to produce the current layer's GRU output, $\mathbf{h}_i$ (Eq. 5). **III.** layer $i$'s router uses this output to select experts and executes standard MoE computation (Eq. 6). Such operation doesn't introduce sequence-level recurrence and can be efficiently implemented, as shown in Tab. 1 and Tab. 3.

et al. (2023) demonstrates that upcycling a standard T5-base (248M) into its MoE counterpart (2B) by copying existing FFN can bring some improvements, but it still lags behind the T5-large with 783M parameters. Similarly, Dai et al. (2024) use fine-grained and shared experts to improve the effectiveness, but the 16B MoE performs comparably with the 7B standard model (Bi et al., 2024).

One potential bottleneck for the current MoE could be the router. Typically, the router is parameterized as one lightweight linear layers, which may limit its capacity to explore the optimal token-expert combination. Previous works also reveal such limitations. For instance, Xue et al. (2024) finds the routing results converge to the token-id-based routing very quickly during the early phase of pre-training, which means the token-expert combination is far from well-explored. Some works even show hash functions (Roller et al., 2021), stochastic routing policy (Zuo et al., 2021), and fixed-random router (Chen et al., 2023) achieves competitive performance with the learnable router, illustrating that the learnable router component in MoE needs further enhancement.

Despite some enhancements for router (Chi et al., 2022; Shen et al., 2023; Do et al., 2023; Chen et al., 2023), current routers in different MoE layers still operate independently without comprehensive investigations into the decisions of other layers. This isolation may lead to suboptimal expert utilization, as each layer manages its routing based solely on local information, potentially leading to inefficiency of model parameters. Though vanilla MoE models could technically share the routing information via hidden states residual, this information may be overshadowed by the language modelling loss, requiring routing-relevant information to "compete" for its representation.

To this end, we introduce a dedicated component to capture and pass routing information for each layer. The proposed architecture, **R**ecurrent **R**outer for **M**ixture-**o**f-**E**xperts (RMoE), is shown in the Fig. 1. Concretely, we regard routing decisions in consecutive layers as a sequence in which the routing results of the $i$-th layer should be conditioned on previous layers' decisions. We thus introduce a lightweight Gated Recurrent Unit (GRU) (Dey & Salem, 2017) to capture this dependence and simulate the information flow between routers across layers. Intuitively, GRU has a reset and an update gate to control the information flow across time steps. Hence, such layerwise recurrence will inform the router to which experts the current token was assigned in previous layers, potentially supporting cross-layer collaborations. Furthermore, the introduced GRU is especially for routing. It thus helps to disentangle the states relevant to model prediction and routing decisions.

We validate RMoE's performance with various model sizes, architectures, datasets, and training settings (per-training and supervised fine-tuning), demonstrating that RMoE outperforms a range of baselines. Moreover, RMoE's introduction of a novel computation stage during routing makes it orthogonal to and compatible with most existing methods. We further analyze RMoE and elucidate the primary contributors to its improvement. Our findings indicate that while the GRU in RMoE shares essential cross-layer information, it also enables additional gradient propagation for the router. Our analysis shows that layerwise recurrence provides cross-layer information, fostering router exploration and optimizing expert utilization. Consequently, the selected experts are leveraged more effectively, leading to increased diversity of experts. We believe that our innovative router design and massive analysis can offer insights into the development of future MoE models.

## 2 RELATED WORKS: VARIOUS ROUTERS FOR MoE

In this section, we review previous approaches to improve router design in SMoE. For example, XMoE (Chi et al., 2022) first projects hidden states into a lower-dimension space and computes their cosine-similarity to low-dimension expert embeddings, which can prevent the hidden states from collapsing to a linear combination of expert embeddings. Moduleformer (Shen et al., 2023) uses an MLP router with ReLU activation to increase router capacity. SMoE-dropout (Chen et al., 2023) utilizes a fixed random-initialized linear router and gradually increases Top-k during training. HyperMoE (Do et al., 2023) introduces a fixed random-initialized hypernet (Ha et al., 2016) at each layer to generate router weights condition on input and one learnable router embedding. One concurrent work (Gong et al., 2024) also introduces GRU in sequential routing stages. However, it does not view such a recurrent mechanism as a general and composable method with broad MoE fields or provide relative ablation or analysis. Extra discussion of related work to improve MoE from routing and training strategies, and utilize recurrent controllers can be found in App. A.1.

## 3 METHODOLOGY

### 3.1 PRELIMINARIES

**Mixture-of-Experts**  MoEs are typically implemented by replacing transformer models' original feed-forward networks (FFNs) with a group of parallel FFNs and incorporating a router. Suppose there are $N$ experts, denoted as $E_n, n \in [1, N]$. The router $g(\cdot; \mathbf{G}, k)$, defined by its parameters $\mathbf{G} \in \mathbb{R}^{(h,N)}$ and an integer $k$, maps the input $\mathbf{x}$ to a score distribution over the experts, $g(\mathbf{x}; \mathbf{G}, k) \in \mathbb{R}^N$. Given $\mathbf{x} \in \mathbb{R}^h$, the output $\mathbf{y} \in \mathbb{R}^h$ is the weighted sum of the outputs from all experts:

$$\mathbf{y} = \sum_{n \in N} g_n(\mathbf{x}; \mathbf{G}, k) E_n(\mathbf{x}) \tag{1}$$

Typically, $g$ is a simple linear layer followed by a $\mathrm{softmax}$ and a Top-k function. The $n$ th element of $\mathbf{x} \times \mathbf{G} \in R^N$ represents the gating score of expert $E_n$, and the $n$ th column of $\mathbf{G}$ can be regarded as the *expert embedding* for expert $E_n$. When $k$ for Top-k is smaller than $N$, only a subset of experts is involved in the computation, which is known as Sparse Mixture-of-Experts (SMoE) (Shazeer et al., 2017; Fedus et al., 2022).

**Recurrent Neural Networks**  RNNs (Medsker et al., 2001) are designed to handle sequential data by maintaining a hidden state $\mathbf{h}$ that holds the information from previous time steps. This hidden state is updated at each time step $i$ based on the current input $\mathbf{x}'_i$ and the hidden state at the last time step $\mathbf{h}_{i-1}$, formulated as $\mathbf{h}_i = f(\mathbf{h}_{i-1}, \mathbf{x}'_i)$.

The Gated Recurrent Units (GRU) Dey & Salem (2017) module is an advanced variant of RNNs that addresses traditional RNNs' limitations, such as difficulty capturing long-term dependencies and gradient vanishing issues. Given an input $\mathbf{x}'_i$ at time step $i$, GRU first calculates the reset gate $\mathbf{s}_i$ and the update gate $\mathbf{z}_i$ to determine how much of the previous memory to keep and to forget,

$$\mathbf{s}_i = \sigma(\mathbf{W}_s \mathbf{x}'_i + \mathbf{U}_s \mathbf{h}_{i-1}), \qquad \mathbf{z}_i = \sigma(\mathbf{W}_z \mathbf{x}'_i + \mathbf{U}_z \mathbf{h}_{i-1}) \tag{2}$$

where $\sigma$ represented the sigmoid activation function and all $\mathbf{W}$ and $\mathbf{U}$ are tranable parameters. And then, the hidden state $h_t$ is updated by

$$\tilde{\mathbf{h}}_i = \tanh(\mathbf{W}_h \mathbf{x}'_i + \mathbf{s}_i \odot (\mathbf{W}_h \mathbf{h}_{i-1})), \qquad \mathbf{h}_i = (1 - \mathbf{z}_i) \odot \tilde{\mathbf{h}}_i + \mathbf{z}_i \odot \mathbf{h}_{i-1} \tag{3}$$

### 3.2 LAYERWISE RECURRENT ROUTER

Existing routers work independently, this lack of global information may prevent routers from discovering more effective token-expert combinations. Therefore, we integrate a GRU into the routing process, explicitly incorporating historical routing information into the current expert selection for each token. Formally, at the $i$ th layer, we first use a linear layer to project the hidden state $\mathbf{x}_i$ to the dimension of the GRU state $\mathbf{x}'_i \in \mathbb{R}^p$ (usually smaller than the dimension $h$ of $\mathbf{x}_i$. We choose 128 for most of the settings provide further analysis in Tab. 6 and Tab. 7):

$$\mathbf{x}'_i = \mathrm{Proj}_i(\mathbf{x}_i) \tag{4}$$

Importantly, we use separate projectors for each layer since the hidden states $\mathbf{x}$ of different layers vary greatly (more discussion in Sec. 5). This projection output $\mathbf{x}'$, along with the GRU result from the previous layer, $\mathbf{h}_{i-1}$, is then fed into a GRU unit to obtain the current GRU output $\mathbf{h}_i$.

$$\mathbf{h}_i = \text{GRU}(\mathbf{x}'_i, \mathbf{h}_{i-1}). \tag{5}$$

Next, $\mathbf{h}_i$ is input into the router and then expert outputs are aggregated based on the router output:

$$\mathbf{y_i} = \sum_{n \in N} g_n(\mathbf{h}_i; \mathbf{G}_i, k) E_n(\mathbf{x}_i). \tag{6}$$

Here, $\mathbf{y}_i$ represents the output of the $i$-th layer, $\mathbf{h}_i$ is the GRU output, $g_n(\mathbf{h}_i; \mathbf{G_i}, k)$ is the router output computed with routing parameter $\mathbf{G_i}$ in layer $i$. Notice that, unlike traditional RNNs, which use a shared projector together for sequential inputs when the input dimension isn't equal to the RNN's hidden dimension, we use different projectors $\text{Proj}_i$ in Eq. 4 for different layers since hidden states and model weights in different layers usually various a lot (Fig. 11 and Tab. 6).

Despite capturing inter-layer dependencies between routers in different layers, RMoE potentially has other advantages: (1) *Prevent representation collapse*: Chi et al. (2022) identified that the single linear layer routers encourage token embeddings clustering around *expert embedding*, implying a trend toward representation collapse issue. And they propose XMoE to first project hidden states into a low-dimension and then calculate the gating score. Similarly, the projector (Eq. 4) and GRU (Eq. 6) in RMoE also separate hidden states from expert embeddings and can reduce this issue. (2) *Additional Gradient Flow*: Before the inclusion of GRU, the router's gradient mainly derive from the expert weight score $g_n$ in Eq. 1. The introduction of GRU not only provides enriched information about historical routing but also an extra gradient propagation through GRU hidden states. We denote this extra gradient flow as *Recurrent Gradient*, and we empirically demonstrated that this *Recurrent Gradient* is important to RMoE. (3) *Applicable with other MoE design*: the proposed method introduces an additional computation stage into SMoE, it is orthogonal to most existing attempts to improve MoE and is seamlessly compatible with them.

## 4 EXPERIMENTS

### 4.1 EXPERIMENTAL SETTINGS

**Langauge Modeling Tasks and Metrics**   Following (Pham et al., 2024), we first test on two common language modeling tasks: enwiki8 (character-level language modeling, with Bits-Per-Character (**BPC**) as the evaluation metrics) and WikiText-103 (word-level language modeling, with Perplexity (**PPL**) as the evaluation metrics). We employ default train-validation-test splits for each dataset. We report test performances of the best validation checkpoints. More details can be found in App. A.2.

**Configurations and Baselines**   We compare RMoE with other existing router designs. All methods are based on the decoder-only standard switch-transformer architecture with post-norm. Following (Pham et al., 2024), all routers select top-2 experts from 16 experts. Each task is trained on 2 NVIDIA A100 GPUs for about 20 hours. More training configurations can be found in App. A.2. Our baselines include (1) **SMoE**: standard switch-transformers with a standard linear router. (2) **HyperMoE** (Do et al., 2023): the method employs a fixed, randomly initialized hypernetwork (Ha et al., 2016) to produce the weights for the linear router, subsequently allowing the generated linear layer to perform the routing. (3) **SMoE-MLP** (Shen et al., 2023): it replaces the linear router with a two-layer MLP using the GELU activation function. (4) **RandomMoE**: inspired by SMoE-Dropout (Chen et al., 2023) and HyperMoE, we propose to compare with a fixed randomly initialized linear router; this could be a naive baseline for all learnable routers. (5) **XMoE** (Chi et al., 2022): it first down-projects the token embeddings to a lower dimension (default 16) and computes its cosine-similarity with the low-dimension expert embeddings. It also uses a learnable temperature in softmax. (5) **CosineSMoE**, similar to XMoE except without down-projection.

**Pre-Training and SFT paradigm**   As pre-training-then-supervised-fine-tuning has become the standard paradigm, we also evaluate the RMoE in this setting. We conduct preliminary scale-up experiment on a setting of training 0.91B models with 40B tokens. Our pre-training corpus is a multilingual data collection that spans common and specialized domains, including Wikipedia, finance,

Table 1: Performance of RMoE and baselines on two language modeling tasks, Enwiki8 and WikiText-103. **Params** means the non-embedding model parameters and (router parameters). Notice we don't separate unlearnable parameters in HyperMoE and RandomSMoE. **Mem** means the peak GPU memory usage with the same batch-size configurations. **Speed** is the average time for 1k training steps. Results demonstrate that the RMoE outperforms baseline models and achieves comparable memory usage and speed as the standard SMoE.

| Algorithm | Enwiki8 (BPC)↓ | | WikiText (PPL)↓ | | Params | Mem | Speed |
|---|---|---|---|---|---|---|---|
| | val | test | val | test | (M) | (GB) | (s/1k steps) |
| **SMoE** | 1.152 | 1.128 | 31.279 | 33.061 | 36.08 (0.04) | 47.92 | 960.2 |
| **HyperMoE** | 1.162 | 1.139 | 31.918 | 33.374 | 48.41 (12.4) | 49.69 | 962.0 |
| **SMoE-MLP** | 1.164 | 1.137 | 31.430 | 33.142 | 36.79 (0.75) | 48.70 | 964.1 |
| **RandomSMoE** | 1.163 | 1.135 | 31.938 | 33.410 | 36.08 (0.04) | 47.72 | 961.4 |
| **CosineMoE** | 1.148 | 1.122 | 31.367 | 33.047 | 36.08 (0.04) | 48.68 | 962.4 |
| **XMoE** | 1.150 | 1.125 | 31.265 | 32.926 | 36.13 (0.09) | 48.70 | 967.5 |
| **RMoE** | **1.141** | **1.116** | **30.939** | **32.867** | 36.51 (0.47) | 49.46 | 972.9 |

and legal texts. Our model architecture is modified based on Llama family (Touvron et al., 2023). Specifically, we use a 24-layer model and top-4 gating from 16 experts per layer following (Dai et al., 2024). This yields a model with approximately 0.53B activated / 0.91B total parameters. All different routers use the same training configurations. To ensure expert load balance, we employ balance loss with weights 0.01 during training. These experiments are conducted using the Megablocks (Gale et al., 2023) on 8 NVIDIA A100 GPUs for about 5 days. More details can be found in App. A.2. After pertaining, we perform supervised fine-tuning (sft). All models are trained on Alpaca (Taori et al., 2023) with the same configuration. We use lm-evaluation-harness[1] to evaluate the fine-tuned model. To simulate the real LLMs application scenario, we don't perform task-specific fine-tuning and evaluation. Since the models are largely under-trained, they give almost random-guessing results on challenging tasks like MMLU (Hendrycks et al., 2020). Therefore, we only test on tasks (ARC-easy, Hellaswag, PIQA, SciQ, LAMBADA) in lm-evaluation-harness. More details about sft configurations and tasks can be found in App. A.2. We further justify the scalability of RMoE on the setting of training 15B activate 2.7B models with 120B / 400B tokens. Given our utilization of a high-quality pre-training corpus, pre-training on 400B tokens yields better results compared to experimental MoE like OpenMoE (Xue et al., 2024). We find RMoE consistently provides over a one-point improvement in performance on benchmarks such as MMLU, GSM8K, and HumanEval. More details can be found in App. A.3.

## 4.2 Main Results

Tab. 1 shows the performance of RMoE and selected baselines on two language modelling tasks. Our observations are as follows: (1) RMoE performs best on validation and test sets of two tasks, and the recurrent routing mechanism and the introduction of extra GRU block do not severely impact the training speed and memory usage, making RMoE more practical. (2) Comparing SMoE-MLP and SMoE, we find that replacing the original simple linear layer with a more capable MLP does not improve perfor-

Table 2: Performance of combining layer-wise recurrent routing mechanism with XMoE.

| Algorithm | Enwiki8 (BPC)↓ | | WikiText (PPL)↓ | |
|---|---|---|---|---|
| | Val | test | val | test |
| XMoE (8) | 1.160 | 1.132 | 31.74 | 33.55 |
| + GRU router | 1.150 | 1.124 | 31.34 | 32.99 |
| XMoE (16) | 1.150 | 1.125 | 31.27 | 32.93 |
| + GRU router | 1.144 | 1.119 | 31.15 | 32.47 |
| XMoE (32) | 1.140 | 1.114 | 31.30 | 32.71 |
| + GRU router | **1.136** | **1.112** | **31.25** | **32.55** |

mance. It even underperforms the fixed random routing (RandomMoE) on Enwikik8, suggesting that naively increasing model capacity can't result in a more powerful router. Furthermore, *since RMoE introduces novel computation stages in routing and is orthogonal to most existing router designs, it can easily be combined with them.* Tab. 2 showcases the performance of the original XMoE and XMoE with GRU router in different XMoE lower dimensions (8, 16, and 32). We observe that the GRU router benefits all of the 3 configurations of XMoE.

While previous work on improving routers has not mostly been evaluated on large-scale pretraining (Dai et al., 2022; Chi et al., 2022; Do et al., 2023), we scale up RMoE to billion-level parameters and training tokens. We report SMoE and RMoE's evaluation results (both directly evaluated and evaluated after supervised fine-tuning (sft)) in Tab. 3. Existing works suggest freezing the router during SMoE tuning Zoph et al. (2022); we report SMoE's results under freeze and unfreeze settings. Correspondingly, for RMoE, we freeze the GRU and the linear layer under the freeze setting. From Tab. 3, we can observe that (1) Even in large-scale pre-training that requires more

---

[1]https://github.com/EleutherAI/lm-evaluation-harness

Table 3: SMoE and RMoE's pre-training costs and evaluation results in selected informative lm-evaluation-harness tasks. 'sft' means supervised fine-tuning on the Alpaca dataset. The task names and metrics for short names in the table are: '**ARC-e**' for ARC-Easy, acc; '**Hella**' is for Hellaswag, acc-norm; '**Piqa**' for PIQA, acc-norm; '**Lamb**' for LAMBADA, acc. Each model has approximately 0.53B activated parameters out-of 0.91B parameters. RMoE introduces about 3.5M additional parameters relative to SMoE.

| Algorithm | Training | ARC-e | Hella | Piqa | Sciq | Lamb | Avg↑ |
|---|---|---|---|---|---|---|---|
| **SMoE**

Speed: 48.87 s/step
Mem: 48.00 GB | pre-train 20B tokens | 47.14 | 35.51 | 64.69 | 76.2 | 14.61 | 47.63 |
| | +sft | 50.93 | 35.82 | 65.61 | 74.7 | 17.81 | 48.97 |
| | +sft (freeze router) | 50.59 | 35.78 | 66.32 | 74.7 | 18.18 | 49.11 |
| | pre-train 40B tokens | 52.57 | 40.85 | 67.74 | 83.4 | 26.74 | 54.26 |
| | +sft | 53.70 | 42.07 | 68.61 | 83.5 | 32.80 | 56.13 |
| | +sft (freeze router) | 53.45 | 41.94 | 68.88 | 83.1 | 32.06 | 55.88 |
| **RMoE**

Speed: 49.07 s/step
Mem: 48.69 GB | pre-train 20B tokens | 47.01 | 35.91 | 65.23 | 78.7 | 19.13 | 49.20 |
| | +sft | 48.53 | 36.90 | 66.21 | 79.6 | 24.74 | 51.20 |
| | +sft (freeze router) | 49.24 | 36.79 | 66.16 | 79.7 | 24.32 | 51.24 |
| | pre-train 40B tokens | 51.18 | 41.38 | 67.79 | 83.6 | 32.58 | 55.31 |
| | +sft | 53.20 | 43.05 | 68.55 | 83.8 | 37.16 | 57.15 |
| | +sft (freeze router) | 53.11 | 43.16 | 68.77 | 82.8 | 37.57 | 57.08 |

complex parallel training strategies, RMoE brings negligible wall time and memory cost compared with vanilla SMoE. (2) In comparable settings (e.g., the same number of tokens and with/without sft), RMoE outperforms SMoE, and even the best results of SMoE are lower than those of RMoE.

## 5 ABLATION STUDIES

**Which contributes more? More Router parameters or layerwise recurrence.** A straightforward reason for RMoE improvement could be that RMoE introduces additional computation and parameters. To disentangle the effect of introducing more router parameters and layerwise recurrence, we consider the fol-

Table 4: Enwiki8 validation and test BPC for different routing designs. 'NP' stands for not passing recurrent states cross-layer. 'RMoE+NP' has the same parameters and FLOPs as 'RMoE'.

| Algorithm | Val↓ | Test↓ | Paras (M) | Val↓ | Test↓ | Paras (M) |
|---|---|---|---|---|---|---|
| | | Small | | | Medium | |
| SMoE | 1.214 | 1.184 | 15.32 | 1.152 | 1.128 | 36.08 |
| SMoE + MLP | 1.214 | 1.183 | 15.73 | 1.164 | 1.137 | 36.79 |
| RMoE + NP | 1.227 | 1.196 | 15.61 | 1.150 | 1.123 | 36.51 |
| RMoE | 1.213 | 1.183 | 15.61 | 1.141 | 1.116 | 36.51 |

lowing two extra settings: (1) SMoE+MLP: we naively increase the router parameters by replacing the original linear layer with a larger MLP layer; (2) RMoE + NP: we change Eq. 5 to $\mathrm{GRU}(\mathbf{r}_i, \mathbf{h}_0)$ to cancel the layerwise recurrence of RMoE, rendering a stateless GRU. The setting has the same parameters and computation as RMoE. From Tab. 4, we can observe that (1) in our setting, introducing larger routers in SMoE doesn't bring improvement (SMoE v.s. SMoE + MLP). (2) When ablated on the layerwise recurrence in RMoE, the performance largely drops, even worse than SMoE. Both results suggest that the layerwise recurrence is the main contributor.

***Recurrent Gradient* is important to RMoE** Following the aforementioned analysis, we try to further disentangle the effect of the layerwise recurrence. When removing the layerwise recurrence as in the RMoE + NP setting, we remove two information flows across layers: (1) the forward information about previous routers' decisions and (2) the backward gradient propagation through GRU hidden states in different layers. To compare the two information flows, we investigate the following settings: (1) RMoE + detach $h_{i-1}$: in intermediate stage between RMoE and RMoE-NP. By detaching $h_{i-1}$ to stop its gradient computation in Eq. 5, each GRU cell can only use previous information during feed-forward. (2) RMoE + NP + r-$\alpha$: inspired by Realformer (He et al.,

Table 5: Enwiki8 validation and test BPC. 'detach $h_{i-1}$' means detaching the recurrent hidden states before passing it to the next block. 'r-0.5/1.0' means passing the routing logits of the previous block to the current block. 'detach-r' means detaching the gradient computation of passed logits.

| Algorithm | Val | Test |
|---|---|---|
| SMoE | 1.152 | 1.128 |
| RMoE | 1.141 | 1.116 |
| + NP | 1.150 | 1.123 |
| + detach $h_{i-1}$ | 1.159 | 1.133 |
| + NP + r-0.5 | 1.149 | 1.124 |
| + NP + r-1.0 | 1.150 | 1.124 |
| + NP + r-0.5 + detach-r | 1.157 | 1.133 |
| + NP + r-1.0 + detach-r | 1.152 | 1.126 |

2020) that introduces residual attention score to facilitate attention gradient back-propagation, we investigate an intermediate stage between RMoE and RMoE-NP by adding gating logits residual for the RMoE + NP settings. Concretely, the gating score of $i$-th layer for expert $n$ is

$g_n(\mathbf{h}_i; \mathbf{G}_i, k) + \alpha g_n(\mathbf{h}_{i-1}; \mathbf{G}_{i-1}, k)$. It is a straightforward way to supplement router information across layers based on the NP setting. In our experiments, we set $\alpha$ as 0.5 and 1.0. (3) Moreover, we also test detaching the gradient computation of passed logits ($\mathbf{h}_{i-1} \times \mathbf{G}_{i-1}$), denoted as 'detach-r'. From Tab. 5, RMoE + detach $h_{i-1}$ performs even worse than RMoE-NP, showing that the *Recurrent Gradient* is important. Similarly, 'NP+r0.5' and 'NP+r1.0' are comparable with 'NP', showing that the naive gating score residual can't provide effective cross-layer information. The performance of their detached version largely drops, demonstrating the importance of extra gradient passing.

To further validate the gradient passing hypothesis, we test 'NP' and 'NP-r0.5/1.0' on deeper models. The results are summarized in Fig. 2. As the layer increases, we can observe that (1) RMoE consistently outperforms other settings, and RMoE-NP even lags behind SMoE. The possible reason is, without passing recurrent states, RMoE-NP is similar to SMoE-MLP which simply increases router complexity but doesn't refine the router training. (2) RMoE-NP-r0.5 surpasses RMoE-NP, further emphasizing that SMoE's optimization benefits from the added additional gradient flow for routers. The spirit echoes the principles behind residual network, where residual connection are used to create direct paths for gradient propagation, thereby mitigating gradient vanishing

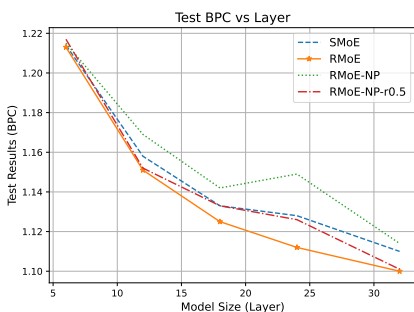

Figure 2: Test BPC on Enwiki8 with different model sizes (6, 12, 18, 24, 32). Similar validation results are in App. A.5 Fig. 14

as lerys deepen. Similarly, the GRU and the direct logits passing help for gradient flow of routers in deep layers. Ad shown in the Fig. 2, as the layer increases. the performance gaps between them may becomes more significant (3) While providing additional gradient across layers, RMoE-NP-r0.5 underperforms RMoE. This may because the indexes of experts in layer $i$ are not aligned with those in other layers, directly adding logits can lead to improper constraints and hurt the model performance, further highlighting that RMoE adds flexible while informative pathways in the SMoE framework.

Table 6: Ablation of RMoE design. 'L-proj' means the layerwise projector in Equ 4, 'S-proj' is the standard RNN projector. 'SMoE + L-proj + GRU router' is our proposed used RMoE method.

| Algorithm | Enwiki8 (BPC)↓ | | WikiText (PPL)↓ | | Params |
| --- | --- | --- | --- | --- | --- |
| | Val | test | val | test | |
| SMoE | 1.152 | 1.128 | 31.28 | 33.06 | 36.08 |
| SMoE + L-proj + GRU router | 1.141 | 1.116 | 30.93 | 32.86 | 36.50 |
| SMoE + S-proj + GRU router | 1.148 | 1.123 | 31.15 | 33.02 | 36.23 |
| SMoE + L-proj + RNN router | 1.145 | 1.119 | 31.18 | 32.72 | 36.44 |
| SMoE + L-proj + LSTM router | 1.148 | 1.122 | 31.19 | 33.04 | 36.54 |

**Layerwise projector and suitable recurrent net bring the best results.** This part tests the other components in RMoE, such as recurrent hidden state dimension, layerwise projector, and GRU cell. As shown in Tab. 6: (1) All methods with recurrent routers outperform SMoE. (2) Layerwise projector in Eq. 5 performs better than standard RNNs using a single shared projector. One possible reason is that the weights and hidden states norm in different layers vary greatly (as shown in App. A.4.5 Fig. 11), and it would be hard for a single shared projector to process them. This approach aligns with the design principle of not sharing LayerNorm parameters when employing shared MoE transformer blocks, as discussed by Xue

Table 7: Ablation of the recurrent design on large scale per-training setting. $p$ is the dimension of the recurrent state $\mathbf{r}_i$ in Eq. 4. We report averaged tasks (the same as Tab. 3) results for pre-trained and stf models. All models are trained with 20B tokens.

| Algorithm | Pretrain | +sft |
| --- | --- | --- |
| SMoE | 47.63 | 49.11 |
| RMoE (GRU, $p = 128$) | 49.20 | 51.32 |
| RMoE (GRU, $p = 256$) | 49.08 | 50.04 |
| RMoE (GRU, $p = 512$) | 49.19 | 50.02 |
| RMoE (RNN, $p = 256$) | 47.92 | 50.44 |

et al. (2022). (3) The GRU router performs best. Moreover, we further compare RMoE variants in the larger scale settings. We compare pre-trained models with different structures and recurrent hidden dimensions in Tab. 7 (Averaged results, full results in App. A.5 Tab. 12). We can find similar results: (1) All RMoE variants outperform SMoE; (2) Simple router (RNN) and complex routers (GRU with $p = 256, 512$) perform worse. In short, *layerwise projector and moderate recurrent cell (e.g. GRU with $p = 128$) effectively introduce layerwise recurrent.*

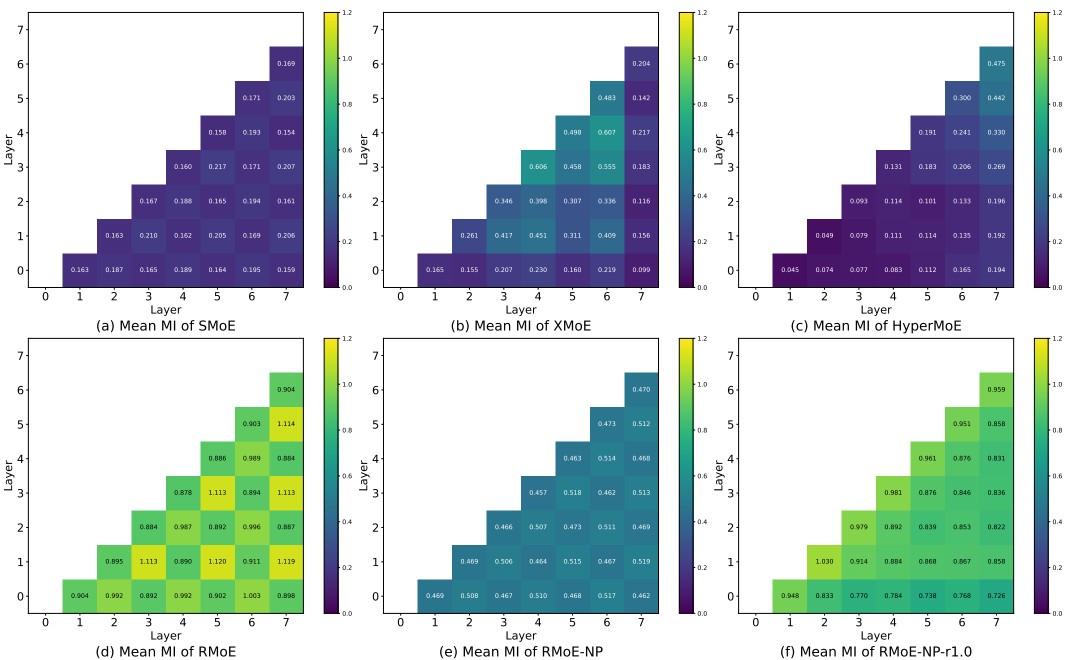

Figure 3: Heat maps of cross-layer mutual information (MI) for different methods. The (i-th row, j-th column) value represents MI between layers i and j. The **First Row** ((a) SMoE, (b) XMoE, (c) HyperMoE): All three methods have low cross-layer MI. **Second Row**((d) RMoE, (e) RMoE-NP, (f) RMoE-NP-r1.0): While RMoE has high cross-layer MI when disabled layerwise recurrent states passing, MI largely drops.

## 6 OBSERVATIONS

**Layerwise recurrence increases cross-layer mutual information.**  The intuition of the proposed RMoE is that current routers in different layers are isolated, and the layerwise GRU is incorporated to provide routers with global information for coordination. Therefore, we measure the Mutual Information (MI) between routing distributions in different layers for each router in Fig. 3. The code can be found in App. A.4.2. We can observe: (1) Besides RMoE, all existing methods show low cross-layer MI, indicating that the routers of different layers work relatively independently. (2) RMoE shows higher MI than three baselines (d v.s. a, b, and c) and RMoE-NP (d v.s. e), showing the recurrent router can facilitate cross-layer information sharing. (3) While RMoE-NP's MI is largely smaller than RMoE, it still surpasses the three baseline methods. The reason can be the shared GRU in Eq. 5. (4) Intuitively, passing routing logits can directly improve MI (f v.s. e). However, directly passing logits can't ensure long-range information sharing, as the values in the right part of (f), which indicate the MI between non-neighbor layers, are smaller than those in (d).

**RMoE enables moderate flat gating scores.**  The router's gating score is a noteworthy feature for MoE-based models. It showcases the models' training dynamics and how they ultimately exploit their experts. Ideally, the training paradigm of MoE models may have two stages: exploration and then exploitation. *i.e.*, the router should actively explore more new expert combinations at the early stage of learning. But if the gating score converges to a sharp distribution too early, the router will learn very shallow routing mechanisms and fail to find optimal routing decisions. So We record gate entropy for each token($-(\sum_n g_n \ln g_n)$, $g_n$ is the gating score for expert $n$) and plot the entropy distribution in Fig. 4 (left). Generally, the higher the entropy, the more evenly the router activates different experts rather than allowing one expert to dominate the layer. Thus, large density in high-entropy parts means many recorded tokens have flat gating score distributions. We can observe that (1) RandomMoE, with a fixed random-initialized router, shows the largest gate entropy. Moreover, most tokens have high entropy, as there is only one peak in the large entropy location. This indicates while RandomRMoE can highly encourage exploration, the router may be under-trained and lack exploitation. (2) SMoE and HyperMoE show low routing entropy, with many tokens having nearly zero entropy. Such low entropy means the softmax operation gives nearly one-hot results, which means the Top-k experts degrade to Top-1 and the router's gradient are very sparse. This can hurt the exploration of expert selection and lead to inefficient Top-k experts usage. (3) XMoE and

CosineMoE, using cosine similarity, which normalized the input and weights $G$ before computing logits, show relatively high entropy. They also perform better than SMoE in Tab. 1, indicating the benefits of suitable exploration. (4) RMoE, with unique cross-layer information sharing, has high entropy for many tokens while low entropy for a few tokens. These moderate gating scores can achieve a better balance between exploration and exploitation.

One may argue that such high entropy may come from the under-trained recurrent router in RMoE instead of capturing the dependency across layers, as the unlearnable RandomMoE also gives high entropy. Therefore, we further visualize the scores of 'RMoE-NP' and 'RMoE-NP-r0.5/1.0' in Fig. 4 (right). The observations are: (1) RMoE-NP's entropy is slightly larger than SMoE's but largely smaller than RMoE's. , indicating that the larger entropy in RMoE is not from under-training but from cross-layer information sharing. (2) While 'RMoE-NP-r0.5' is larger than SMoE and smaller than RMoE, 'RMoE-NP-r1.0' is the largest. From Tab. 5 and Fig. 2, the small and large one both under-perform RMoE, These further demonstrate that the recurrent network can achieve a moderate flat gating score distribution, leading to a better trade-off between exploration and exploitation.

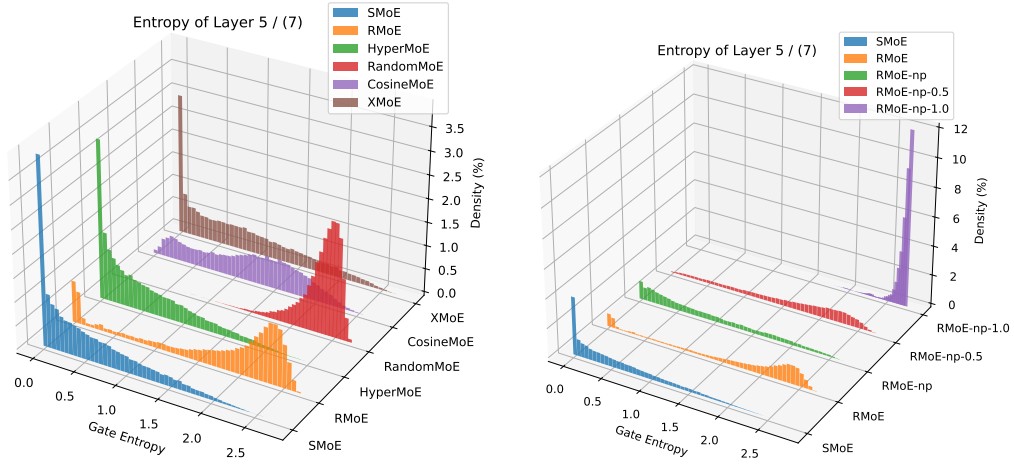

Figure 4: Gate score entropy distribution over Enwiki8 test set for different router configurations. More similar results can be found in App. A.4.4 Fig. 8 and Fig. 9.

We also look into the statistics of selected experts' scores. Here we calculate the (1) **Inner Balance (IB)**: defined as the ratio Top-1 score/Top-2 score, large IB means the first expert dominates all selected experts; and (2) **Outer Balance (OB)**, defined as $\sum_{k \in \text{Top-k}} g_k$, indicating the selected scores' ratio in the score distribution, large OB means selected expert scores dominate the gate score distribution. Because such a ratio could have some extreme values, we report the median number for all tokens in Tab. 8. We can observe: (1) RandomMoE, with a fixed router, shows the lowest IB and OB. (2) Low-entropy models in the previous section (Sec. 6) have high IB and OB. (3) RMoE gives suitable IB and OB. While simply using a complex router ('RMoE-NP') shows relatively low IB and OB, RMoE is even lower. Moreover, passing logits can reduce IB and OB ('RMoE+NP+r-0.5/1.0'). *All these experiments show sharing cross-layer router information can lead to more balanced routing decision and thus facilitate expert usage.*

Table 8: Expert scores balance on Enwiki8. Inner Balance (IB) represents the (top-1 score / top-2 score) ratio, and Outer Balance (OB) represents summed selected gate scores.

| Algorithm | IB | OB |
|---|---|---|
| SMoE | 34.68 | 0.915 |
| HyperMoE | 34.60 | 0.920 |
| CosineMoE | 7.611 | 0.794 |
| XMoE | 19.93 | 0.861 |
| RandomMoE | 2.000 | 0.414 |
| RMoE | 2.021 | 0.573 |
| + NP | 16.58 | 0.842 |
| + NP + r-0.5 | 2.792 | 0.661 |
| + NP + r-1.0 | 1.147 | 0.212 |

**Layerwise recurrence reduces the negative effect of load balance constraint.** To provide a more direct analysis of the router gradient, we investigate how the gradient norm of the router varies throughout the entire training process. When training a MoE model, the gradient of the router has

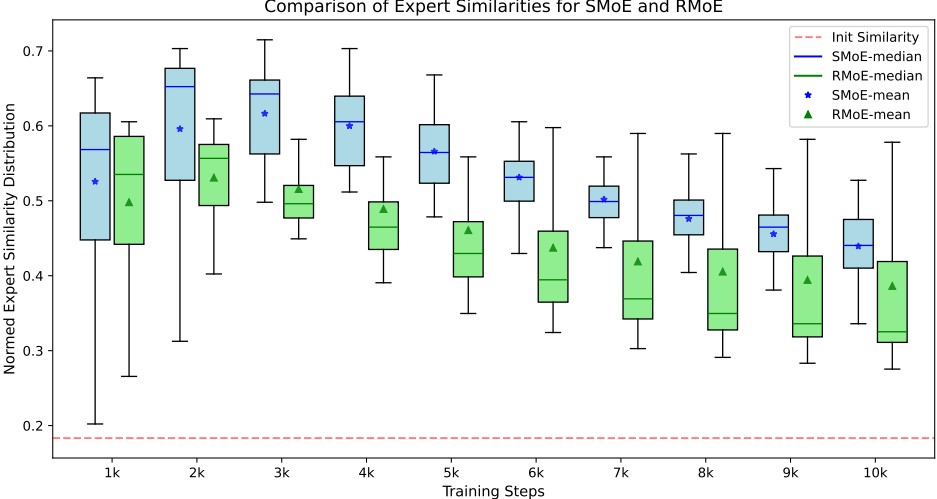

Figure 5: Experts similarity distribution across layers during large-scale pre-training. We plot box plots of expert similarity from checkpoints taken every 1k training steps (approximately 4B tokens), showing the expert similarity across the 24 layers of the model (with maximum, minimum, first quartile, median, and mean).

two separate sources: (1) the language modeling (LM) loss, and (2) the load balancing (LB) loss that pushes the router to assign tokens to different experts in a balanced manner. We empirically find (1) LB loss dominates the training of the linear router at the early training stage. This could hurt model's general performance, as Wang et al. (2024) find, a high LB loss can cause balance token distribution but reduce performance. (2) On the contrary, the gradient of the RNN router from LB loss stabilises in the early stage, and the gradient from the LM loss keeps decreasing, suggesting that the RNN router is more optimised towards the LM loss. These observations suggest the recurrent router can effectively controls the influence of the LB loss. More details can be found in App. A.4.1

**Layerwise recurrence encourages expert diversity** One intriguing feature of MoE is that experts could modularly specialize on different inputs. Therefore, following recent works that analyze the FFNs (Geva et al., 2021; Qiu et al., 2024a;b) and expert weights similarity (Wu et al., 2022; Lo et al., 2024), we use the cosine-similarity of expert's parameters to measure the expert diversity . We calculate for SMoE and RMoE in the large-scale pre-training settings, and the results are shown inFig. 5. To better understand the scale of similarity score, we also plot one dash line showing the similarity of random initialized experts. More details about similarity calculation and explanation can be found in App. A.4.3. We can observe that: (1) At the beginning of the training, the lowest expert similarities are similar to the random initialized one. (2) The expert similarity increases in the early training stages, then decreases later. This may be due to the randomly initialized router in the early stages, which essentially assigns tokens randomly to different experts, leading to increased expert similarity. As the router continues to learn, it gradually assigns specific tokens to the corresponding experts, resulting in decreased expert similarity as training progresses. (3) During the entire training stages, the average similarity score between experts in RMoE is lower than those in SMoE, indicating that RMoE encourages more diverse experts. This expert diversity also reasonably corresponds to the moderate flat gate scores in Sec 6.

## 7 CONCLUSION

This work introduces a layer-wise recurrent router for existing MoE-based language models. We validate the effectiveness of this layer-wise recurrence across various settings, tasks, and model sizes. By adding a new yet efficient computation stage in the routing, RMoE stands orthogonal to most existing methods and can be flexibly integrated with them. Ablation studies reveal that this recurrent mechanism offers additional *Recurrent Gradients*, aiding router optimization. Further analysis validates our intuition that GRU facilitates inter-layer information sharing. We also systematically compare RMoE's model behavior with various baseline models, demonstrating that RMoE can enhance existing SMoE methods and providing insights for future research.

ACKNOWLEDGMENT

Jie Fu is supported by Shanghai Artificial Intelligence Laboratory.

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

# A  APPENDIX

## A.1  MORE RELATED WORKS

**Routing Strategies**  While most MoE works follow the original success and use token choice routing, some works explore different routing approaches. In Expert-Choice Routing (Zhou et al., 2022), each expert selects tokens to process across the whole batch input. This method avoids expert imbalance issues and allows different tokens to be processed by a flexible number of experts. Soft Mixture-of-Experts (Puigcerver et al., 2023) further assigns token weights for input tokens, weighted-averages them, and passes these merged tokens to different experts. This method moves one step behind the Expert-Choice Routing to allow more precise control. However, their token-selecting operations are non-causal and thus can't be directly used in the decoder models. Recent works (Huang et al., 2024; Yang et al., 2024) introduce dynamic top-k for each input token. While the FLOPs can be reduced, since this dynamic assignment can hurt the parallel computation of experts, more system-level implementation must be optimized to achieve wall-time efficiency. Some works also analyze issues in the routing of standard MoE like uncertain tokens (Wu et al., 2024) and lack of expert knowledge transfer (Zhao et al., 2024).

**Training Strategies**  Due to the unstable nature of MoE (Zoph et al., 2022), some works investigate special training strategies for MoE. EvoMoE (Nie et al., 2021) uses a large top-$k$ (even equal to the expert number) at the beginning of training, gradually decreasing $k$. StableMoE (Dai et al., 2022) proposes to freeze the router after training some tokens to avoid token assignment conflicts. Residual Mixture of Experts (Wu et al., 2022) initializes MoE from dense training checkpoints and finds it is an efficient method to train MoE models. Later, sparse-upcycling (Komatsuzaki et al., 2023) further trains large-scale language models from dense checkpoints, and many works follow this paradigm to efficiently utilize the power of MoE in fine tuning (Li et al., 2023), instruction tuning (Lin et al., 2024), and visual instruction tuning (Ding et al., 2024). Different from directly training MoE models, some works continue training the same pre-trained model on several different datasets to encourage specialization and combine them, either merging them into an MoE-style model (Gururangan et al., 2021; Sukhbaatar et al., 2024) or keeping a group of models and introducing a model-level router (Li et al., 2022; Gururangan et al., 2023).

**Recurrence Controller**  A series of works introduce recurrent networks for Neural Architecture Search (NAS) (Zoph & Le, 2016; Ramachandran et al., 2017; Pham et al., 2018; Liu et al., 2018). They introduce a recurrent controller network that predicts the current layer-$i$'s architecture (like CNN filters' number, size, and stride) based on layer-$i$'s input hidden states and previous recurrent states (Zoph & Le, 2016). While these works use RNN to predict model architecture configurations of each layer for all inputs, RMoE utilizes RNN to help the router select expert combinations for each token, which can be viewed as a dynamic version of NAS.

## A.2  EXPERIMENT SETUP

**Enwiki8 and WikiText-103**  We follow the default configurations in CompeteSMoE (Pham et al., 2024). Each model is trained for 80,000 steps with Adam optimizer. The learning rate is 0.0007 with 4000 warmup steps, and the batch size is 48. The main used model is a decoder-only transformer-based architecture with 8 layers and a hidden size of 352. It includes 16 experts, where the top 2 are selected during computation, each with an expert size of 352. The model uses 8 attention heads and handles sequences up to 512 tokens in length, with an attention span of 2048 tokens. It incorporates a dropout rate of 0.1 and a load balancing factor of 0.01 to ensure an even distribution of expert utilization. **Computation Cost** Each 8-layer model is trained on one NVIDIA-A100 GPU for approximately 21 hours.

**Large Scale Pre-training**  For model architecture, our 24-layer model employs Rotary Embedding for positional encoding, SwiGLU for activation functions, and RMSNorm to enhance the model's efficiency and performance. Other model configuration includes a hidden size of 1280, 20 attention heads, an initialization method standard deviation of 0.02, a sequence length of 4096, and a maximum positional embedding length of 4096. All dropout rates are set to 0. For the MoE part, we use 16 experts, with each expert having a feedforward network hidden size of 448, following

the fine-grained MoE settings, and each token activating 4 experts. We use a tokenizer with a 96512 vocabulary size, which adds approximately 123M embedding parameters and 123M vocabulary projection head parameters. Under this configuration, each model has approximately 664M non-embedding parameters, and every token activates 334M non-embedding parameters. The total parameter is around 910M. **For pre-training configurations**, we use a global batch size of 1120, a warmup period of 2000 iterations, a learning rate of 4.2e-4, a minimum learning rate of 4.2e-5, cosine learning rate decay, Adam optimizer with $\beta_1 = 0.9$ and $\beta_2 = 0.95$, a weight decay of 0.1, and gradient clipping at 1.0. **Computation Cost** Each 24-layer model is trained on 8 NVIDIA-A100 GPUs for approximately 5 days.

**Instruction Tuning Data** The **Alpaca** (Taori et al., 2023) dataset is an open-source instruction-following dataset created by Stanford researchers, inspired by OpenAI's ChatGPT. The dataset consists of 52,000 instruction-response pairs generated using the text-davinci-003 model by providing diverse and comprehensive instructions and recording the corresponding responses. It is designed to facilitate the training and evaluation of models in understanding and generating human-like text responses to various instructions.

**Instruction Tuning Setting** We use the codebase[2] and corresponding default configurations. More concretely, we use bfloat16 (bf16) precision to accelerate training while maintaining numerical stability. The model is trained for 3 epochs using AdamW optimizer with a global batch size 128. We set the learning rate to 2e-5 and do not apply weight decay. A warmup ratio of 0.03 is used to gradually increase the learning rate at the beginning of training, and we utilize a cosine learning rate scheduler to adjust it throughout the training process, promoting smoother convergence. **Computation Cost** Each is trained on 8 NVIDIA-A100 GPUs for approximately 2 hours.

**Evaluation Tasks** Here we shortly describe our used evaluation datasets:

**ARC-Easy** is a subset of the AI2 Reasoning Challenge (ARC) dataset (Clark et al., 2018). It consists of multiple-choice questions from elementary and middle school science exams that are relatively easier than the ARC-Challenge set. These questions require basic reasoning and knowledge application.

**Hellaswag** (Zellers et al., 2019) is a dataset designed for commonsense reasoning and narrative prediction. It involves choosing the most plausible continuation of a given scenario from multiple options. The task is challenging because it requires understanding and applying common sense knowledge.

**PIQA** (Bisk et al., 2020) dataset tests a model's ability to understand and reason about physical interactions and affordances. The task involves selecting the correct answer to questions about everyday physical activities.

**SciQ** (Welbl et al., 2017) is a dataset of science questions that includes multiple-choice and direct-answer formats. It aims to test a model's ability to understand and reason with scientific concepts typically taught at the school level.

**LAMBADA** (Paperno et al., 2016) is a dataset designed for language modeling and comprehension. The task involves predicting the last word of a given passage, which requires a deep understanding of the context provided by the preceding text.

A.3    FURTHER PRETRAINING VALIDATION

To further validate the scalability of RMoE, we conduct experiments with larger model sizes and increased pre-training corpus. Both MoE models followed the design principles of DeepSeek-MoE (Dai et al., 2024), utilizing fine-grained experts and shared experts to maintain strong baselines. We evaluated the models on more challenging benchmarks, including Hellaswag, MMLU, GSM8K, and HumanEval, to assess their language capabilities, multi-domain knowledge, mathematical skills, and coding abilities. Additionally, we tested the models' perplexity on multiple domain test datasets and reported the average results.

---

[2]https://github.com/tatsu-lab/stanford_alpaca

Tab. 9 and Tab. 10 present the performance of a 15-billion parameter model with 2.7 billion activated experts, trained on 120 billion and 400 billion tokens, respectively. The results show that RMoE consistently delivers improvements even with increased data volumes. The findings indicate that RMoE enhances performance in standard language modeling tasks, such as Hellaswag and PPL, and improves performance on more complex reasoning tasks.

Table 9: Performance comparison of SMoE, SMoE-MLP and RMoE at the model scale of 15B activation 2.7B parameters, training 120B tokens.

| | Hellaswag | MMLU | GSM8K | Avg PPL |
|---|---|---|---|---|
| Pretrain 80B Tokens | | | | |
| SMoE | 67.69 | 46.24 | 24.18 | 7.406 |
| SMoE-MLP | 67.98 | 46.47 | 23.58 | 7.437 |
| RMoE | 68.00 | 47.74 | 27.14 | 7.361 |
| Pretrain 100B Tokens | | | | |
| SMoE | 70.98 | 50.61 | 30.78 | 6.754 |
| SMoE-MLP | 70.8 | 50.6 | 30.17 | 6.786 |
| RMoE | 71.02 | 51.74 | 32.98 | 6.732 |
| Pretrain 120B Tokens | | | | |
| SMoE | 72.03 | 52.79 | 34.8 | 6.447 |
| SMoE-MLP | 72.19 | 52.81 | 34.57 | 6.479 |
| RMoE | 72.36 | 54.02 | 36.13 | 6.425 |

Table 10: Performance comparison of SMoE, SMoE-MLP and RMoE at the model scale of 15B activation 2.7B parameters, training 400B tokens.

| | Hellaswag | MMLU | GSM8K | Avg PPL |
|---|---|---|---|---|
| Pretrain 200B Tokens | | | | |
| SMoE | 69.48 | 49.96 | 33.21 | 7.718 |
| SMoE-MLP | 69.76 | 50.27 | 31.77 | 7.736 |
| RMoE | 70.00 | 52.21 | 32.98 | 7.608 |
| Pretrain 280B Tokens | | | | |
| SMoE | 72.40 | 54.66 | 42.61 | 6.477 |
| SMoE-MLP | 72.62 | 55.33 | 38.51 | 6.502 |
| RMoE | 73.18 | 56.06 | 44.35 | 6.400 |
| Pretrain 400B Tokens | | | | |
| SMoE | 76.39 | 59.54 | 52.16 | 5.685 |
| SMoE-MLP | 76.09 | 59.96 | 51.71 | 5.709 |
| RMoE | 76.72 | 60.60 | 52.99 | 5.620 |

## A.4 ADDITIONAL OBSERVATIONS

### A.4.1 Router Gradient Norm and Drop Ratio

Table 11: Comparison of linear and RNN routers in terms of gradients and drop ratios at various training steps. We record the router gradient every 10k training steps (20B tokens). We compute the gradient with language modeling (LM) loss and load balance (LB) loss. Drop ratio is the ratio of dropped tokens and all tokens as we assign capacity factor 1.0 for each expert.

| Training steps (k step) | 0.1 | 10 | 20 | 30 | 40 | 50 | 60 |
|---|---|---|---|---|---|---|---|
| Linear router | | | | | | | |
| grad from the whole loss | 1.058 | 0.194 | 0.1911 | 0.198 | 0.208 | 0.217 | 0.221 |
| grad from LM loss | 0.625 | 0.183 | 0.184 | 0.192 | 0.204 | 0.215 | 0.220 |
| grad from LB loss | 0.433 | 0.011 | 0.008 | 0.006 | 0.004 | 0.002 | 0.001 |
| drop ratio | 35.6 | 5.43 | 5.34 | 5.17 | 4.89 | 4.64 | 4.50 |
| RNN router | | | | | | | |
| grad from the whole loss | 0.972 | 0.160 | 0.153 | 0.153 | 0.155 | 0.155 | 0.154 |
| grad from LM loss | 0.636 | 0.146 | 0.138 | 0.139 | 0.144 | 0.148 | 0.151 |
| grad from LB loss | 0.337 | 0.014 | 0.015 | 0.014 | 0.011 | 0.007 | 0.003 |
| drop ratio | 38.7 | 6.35 | 6.30 | 5.94 | 5.32 | 4.54 | 4.09 |

Based on the setting of training 15B models for 120B tokens, we investigate how the gradient norm of the router varies throughout the entire training process. When training an MoE-based model, the gradient of the router has two separate sources: due to (1) the language modeling (LM) loss, and (2) the load balancing (LB) loss that forces the router to assign tokens to different experts in a balanced manner. Therefore, for each router, we compare the gradient from the LM loss only and from the whole training loss. We calculate the average for 100 training steps to estimate the gradient norm.

Furthermore, to better investigate the relation between the router behavior and the router gradient, we calculate the **drop ratio** for the router. This is because during the large-scale MoE pre-training, to ensure the training efficiency, the expert is usually controlled by an hyper-parameter called capacity factor, which determines the total tokens that one expert can process. If the router assigns tokens to some expert that exceeds its capacity, the expert will drop tokens with the lowest scores. And we define the drop ratio as tokens dropped / total tokens. The LB loss mentioned before is critical to decreasing the drop ratio.

According to Tab. 11, we have the following observations: 1. The gradient norm of the RNN router is generally smaller than that of the linear router. And for both routers, the drop ratio decreases with the training. 2. According to the drop ratios, we observe the significant behavioral difference between the two routers: during the early training phase (10k steps -¿ 30k steps), the drop ratio of the linear router is noticeably lower than that of the RNN router; the drop ratio of the RNN router archives at the lower value in the end. 3. The trend observed in the drop ratio is consistent with the results of the gradient norm. The grad norm for LB loss is relatively higher in the RNN router until the final training stage (50k - 60k), whereas the gradient from LB loss in the linear router is high at the beginning and generally low during the later part of training (10k - 60k).

These phenomena indicate that the LB loss could dominate the training of the linear router: when the drop ratio is low and stays unchanged, the grad from LB loss will be low because the router is already well-optimized for LB loss. Such early convergence in the LB loss may reach a suboptimal solution in the trade-off between optimizing load balance and language modeling. On the contrary, the gradient of the RNN router from LB loss stabilizes in the early training steps (10k - 30k), and the gradient from the lm loss keeps decreasing, suggesting that the RNN router is more optimized towards the LM loss.

### A.4.2 MUTUAL INFORMATION

```python
import numpy as np
from sklearn.metrics import mutual_info_score

def discretize_prob_dist(prob_dist, bins=100):
    """
    Discretize the probability distribution into discrete bins.
    """
    discretized = np.digitize(prob_dist, bins=np.linspace(0, 1, bins))
    return discretized

def calculate_mutual_information(x1, x2, bins=100):
    """
    Calculate mutual information between each pair of distributions in x1 and x2.
    x1, x2: numpy arrays of shape (N, 16)
    bins: number of bins to use for discretization
    Returns a numpy array of mutual information values.
    """
    mi_values = []
    for i in range(x1.shape[0]):
        x1_discretized = discretize_prob_dist(x1[i], bins)
        x2_discretized = discretize_prob_dist(x2[i], bins)
        mi = mutual_info_score(x1_discretized, x2_discretized)
        mi_values.append(mi)
    return np.array(mi_values)
```

### A.4.3 EXPERT SIMILARITIES

```python
def get_similarities(htoh4_0, htoh4_1, h4toh):
    avg_key_0 = htoh4_0.mean(dim=1) # (num_experts, 4h, h)
    avg_key_1 = htoh4_1.mean(dim=1) # (num_experts, 4h, h)
    avg_value = h4toh.mean(dim=2) # (num_experts, h, 4h)
    normed_key_0 = nn.functional.normalize(avg_key_0, p=2, dim=1)
    normed_key_1 = nn.functional.normalize(avg_key_1, p=2, dim=1)
    normed_value = nn.functional.normalize(avg_value, p=2, dim=1)
    normed_avg_expert = torch.cat([normed_key_0, normed_key_1, normed_value], dim=1)
    # compute the average expert similarity
    similarity = torch.mm(normed_avg_expert, normed_avg_expert.t())
    avg_sim = normed_similarity.mean().item()
    return  avg_sim
```

### A.4.4 MORE ROUTER ENTROPY DISTRIBUTIONS

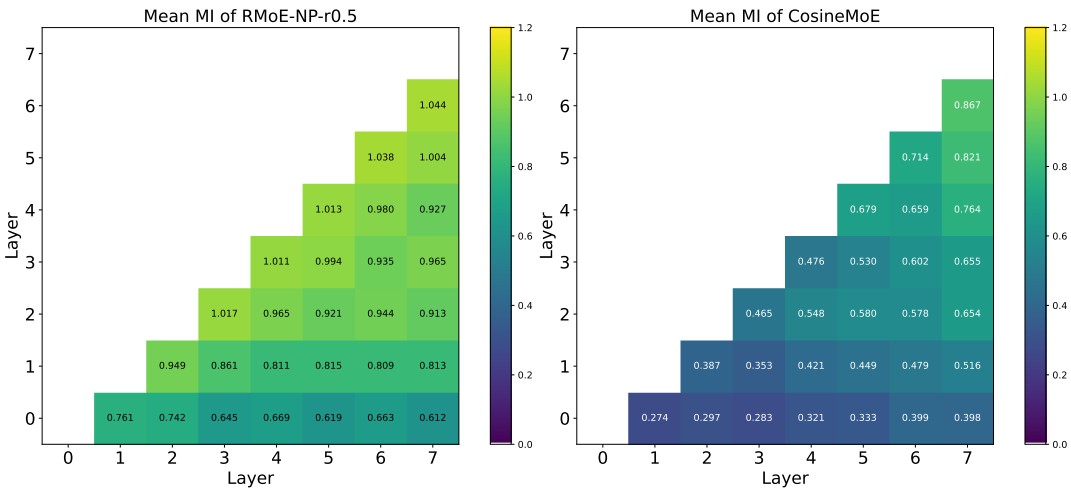

Figure 6: Mutual information of RMoE-NP-r0.5 and CosineMoE settings

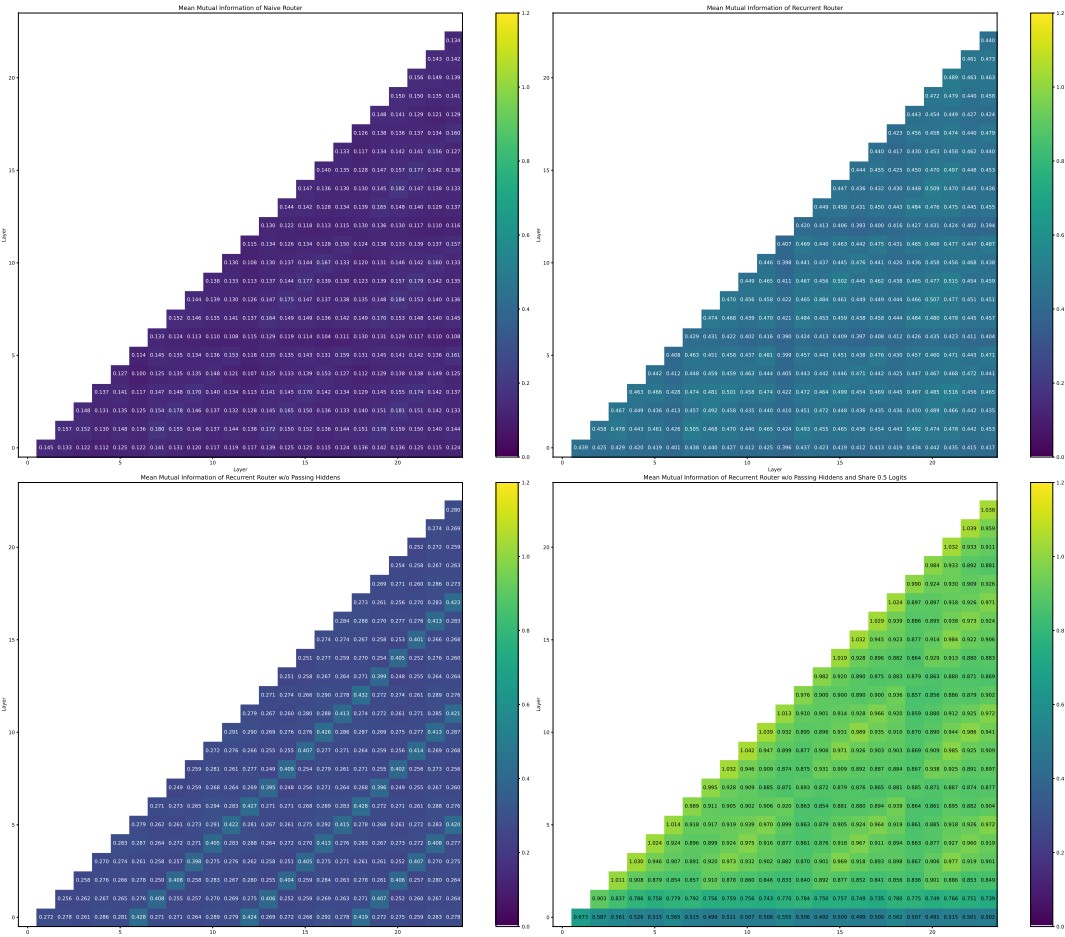

Figure 7: Mutual information of SMoE, RMoE, RMoE-NP, and RMoE-NP-r0.5 in 24-layer models.

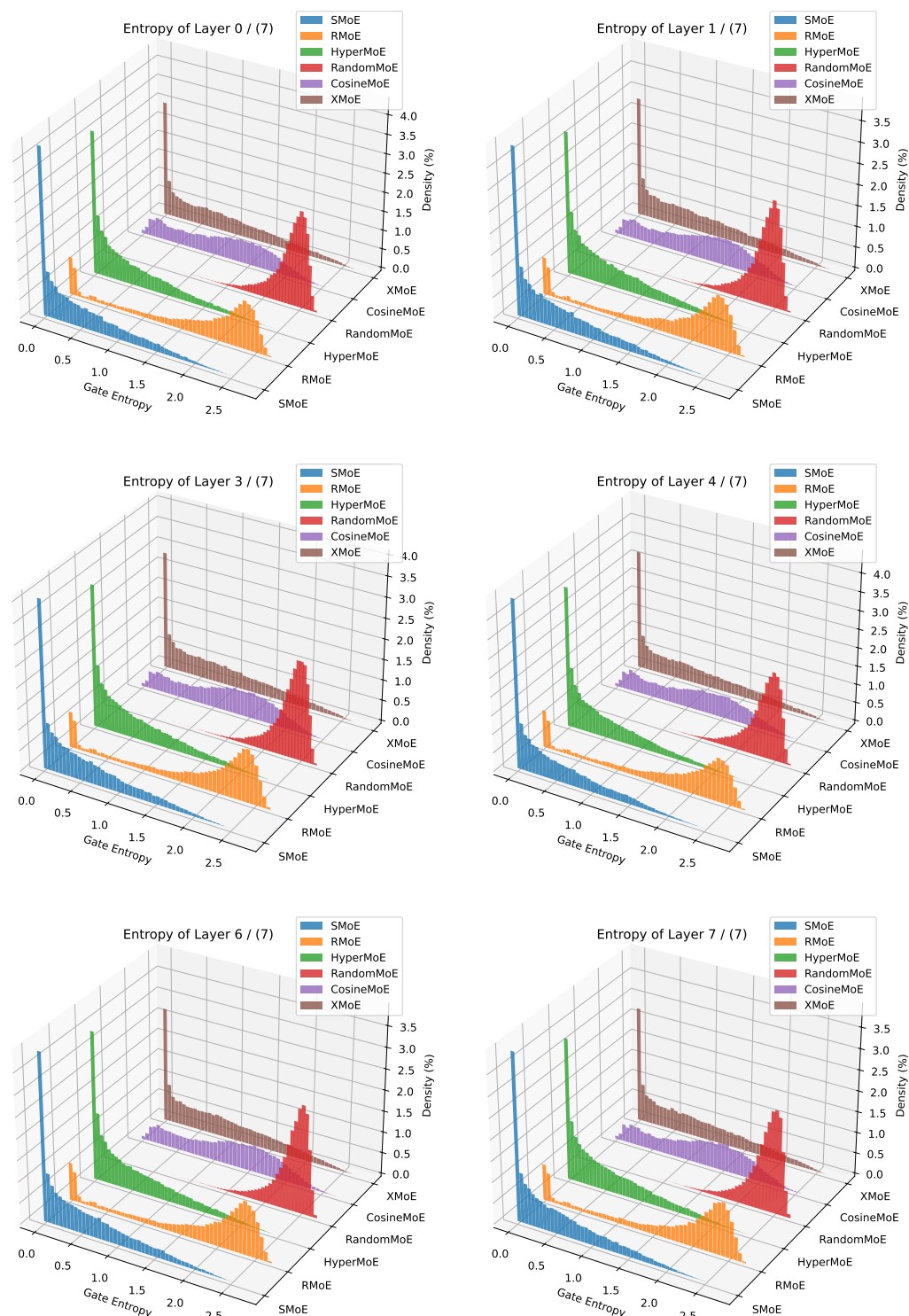

Figure 8: Gate score entropy distribution over Enwiki test set for different routers in 8-layer models.

### A.4.5 ROUTER WEIGHTS INFORMATION

### A.4.6 EXPERT SELECTION FREQUENCY

### A.5 ADDITIONAL RESULTS

Table 12: More SMoE and RMoE variants pre-training costs and evaluation results in selected informative lm-evaluation-harness tasks. 'sft' means supervised fine-tuning on the Alpaca dataset. The task names and metrics for short names in the table are: '**ARC-e**' for ARC-Easy, acc; '**Hella**' is for Hellaswag, acc-norm; '**Piqa**' for PIQA, acc-norm; '**Lamb**' for LAMBADA, acc.

| Algorithm | Training | ARC-e | Hella | Piqa | Sciq | Lamb | Avg↑ |
|---|---|---|---|---|---|---|---|
| **SMoE** | 20B (5k steps) | 47.14 | 35.51 | 64.69 | 76.2 | 14.61 | 47.63 |
| | +sft | 50.93 | 35.82 | 65.61 | 74.7 | 17.81 | 48.97 |
| | +sft (freeze gate) | 50.59 | 35.78 | 66.32 | 74.7 | 18.18 | 49.11 |
| | 40B (10k steps) | 52.57 | 40.85 | 67.74 | 83.4 | 26.74 | 54.26 |
| | +sft | 53.7 | 42.07 | 68.61 | 83.5 | 32.8 | 56.13 |
| | +sft (freeze gate) | 53.45 | 41.94 | 68.88 | 83.1 | 32.06 | 55.89 |
| **RMoE** GRU $p = 128$ | 20B | 47.01 | 35.91 | 65.23 | 78.7 | 19.13 | 49.20 |
| | +sft | 48.53 | 36.9 | 66.21 | 79.6 | 24.74 | 51.20 |
| | +sft (freeze router) | 48.65 | 36.88 | 66.43 | 80.1 | 24.55 | 51.32 |
| | +sft (freeze router and GRU) | 49.24 | 36.79 | 66.16 | 79.7 | 24.32 | 51.24 |
| | 40B | 51.18 | 41.38 | 67.79 | 83.6 | 32.58 | 55.31 |
| | +sft | 53.20 | 43.05 | 68.55 | 83.8 | 37.16 | 57.15 |
| | +sft (freeze router) | 53.03 | 42.96 | 68.34 | 83.6 | 36.68 | 56.92 |
| | +sft (freeze router and GRU) | 53.11 | 43.16 | 68.77 | 82.8 | 37.57 | 57.08 |
| **RMoE** GRU $p = 256$ | 20B | 47.47 | 35.91 | 65.78 | 76.2 | 20.03 | 49.08 |
| | +sft | 48.36 | 36.49 | 65.07 | 77.4 | 22.86 | 50.04 |
| | +sft (freeze router) | 48.27 | 36.42 | 65.23 | 76.9 | 22.88 | 49.94 |
| | +sft (freeze router and GRU) | 48.23 | 36.46 | 64.94 | 77.3 | 22.61 | 49.91 |
| | 40B | 53.07 | 41.15 | 68.52 | 84.0 | 19.17 | 53.18 |
| | +sft | 54.46 | 43.06 | 67.46 | 84.9 | 24.57 | 54.89 |
| | +sft (freeze router) | 54.45 | 43.10 | 67.19 | 84.1 | 23.93 | 54.55 |
| | +sft (freeze router and GRU) | 54.50 | 43.13 | 67.36 | 83.8 | 23.62 | 54.48 |
| **RMoE** GRU $p = 512$ | 20B | 47.77 | 35.39 | 64.80 | 79.5 | 25.00 | 50.49 |
| | +sft | 48.27 | 36.47 | 65.51 | 76.6 | 22.18 | 49.81 |
| | +sft (freeze router) | 47.73 | 36.41 | 65.78 | 76.6 | 22.88 | 49.88 |
| | +sft (freeze router and GRU) | 48.19 | 36.22 | 65.29 | 76.8 | 23.5 | 50.00 |
| | 40B | 51.64 | 41.37 | 66.81 | 86.0 | 22.76 | 53.72 |
| | +sft | 52.82 | 42.68 | 68.55 | 86.0 | 26.88 | 55.39 |
| | +sft (freeze router) | 52.48 | 42.61 | 68.44 | 86.0 | 27.23 | 55.35 |
| | +sft (freeze router and GRU) | 52.74 | 42.44 | 68.77 | 86.3 | 27.13 | 55.48 |
| **RMoE** RNN $p = 256$ | 20B | 46.63 | 35.7 | 64.91 | 76.1 | 16.24 | 47.92 |
| | +sft | 48.40 | 36.45 | 65.51 | 77.3 | 22.65 | 50.06 |
| | +sft (freeze router) | 48.70 | 36.29 | 65.45 | 77.3 | 22.60 | 50.07 |
| | +sft (freeze router and RNN) | 49.24 | 36.48 | 65.56 | 77.7 | 23.20 | 50.44 |

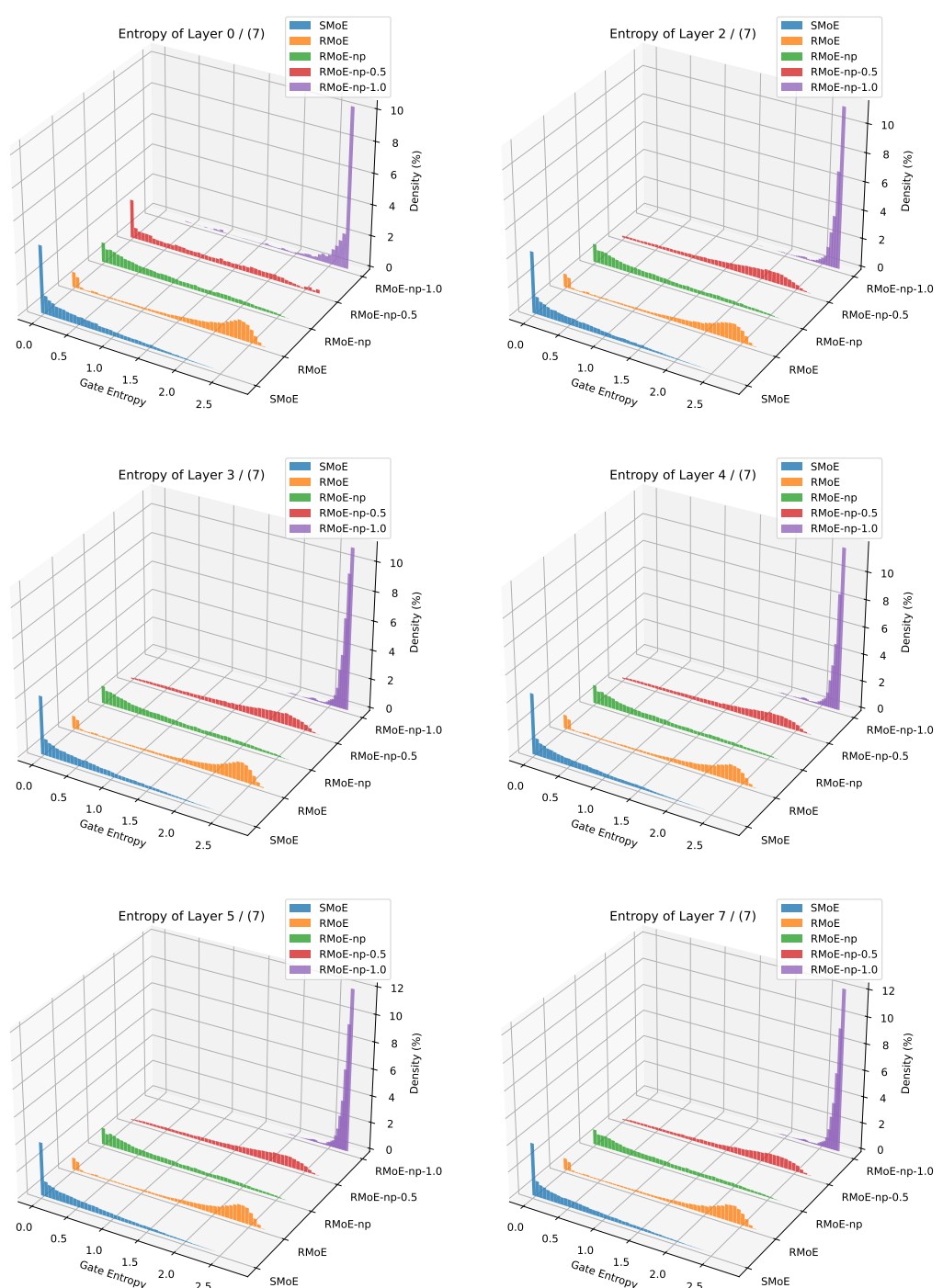

Figure 9: Gate score entropy distribution over Enwiki test set for different information passing settings in 8-layer models.

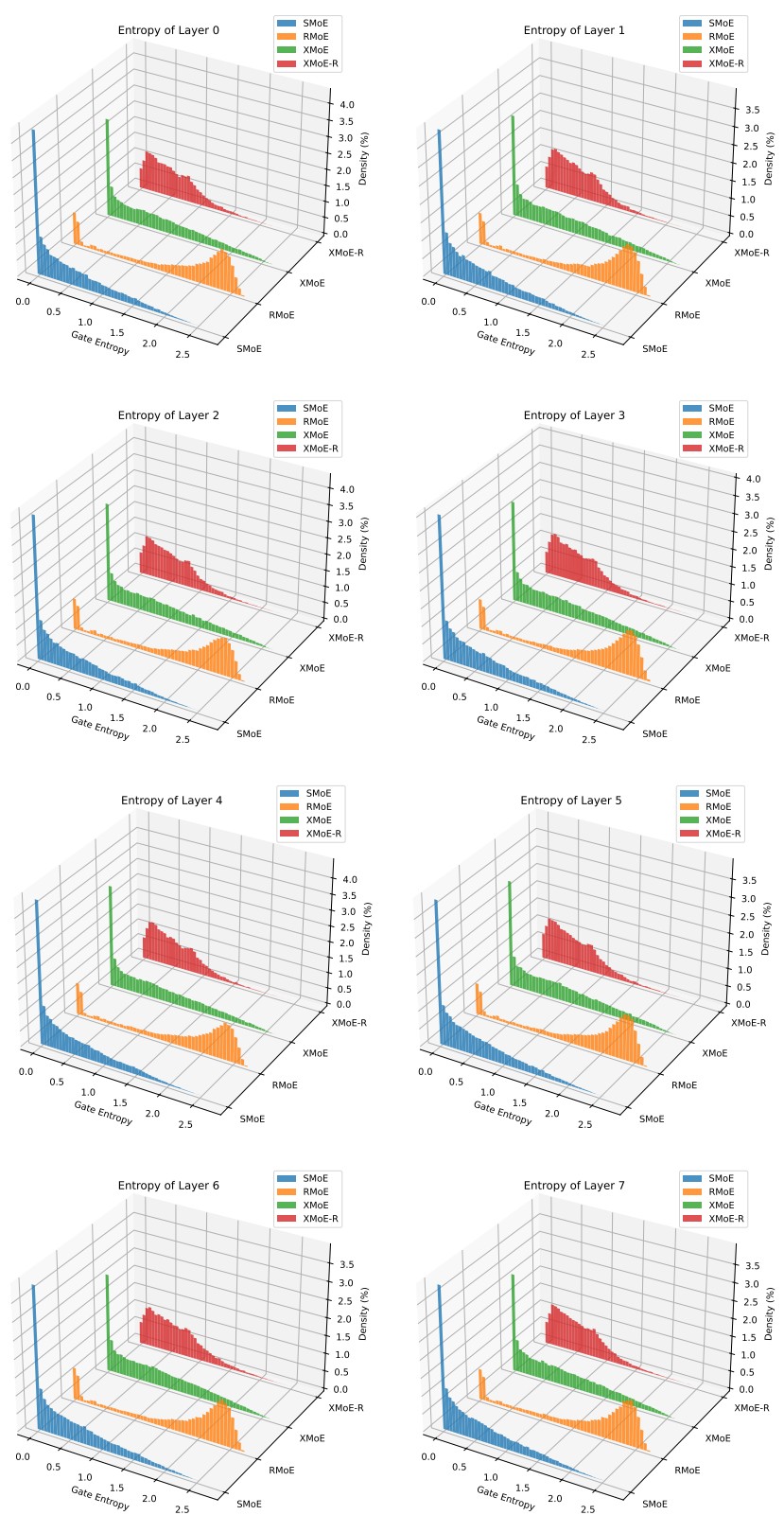

Figure 10: Gate score entropy distribution over Enwiki test set for different routers. RMoE can be combined with XMoE to encourage the exploration of XMoE.

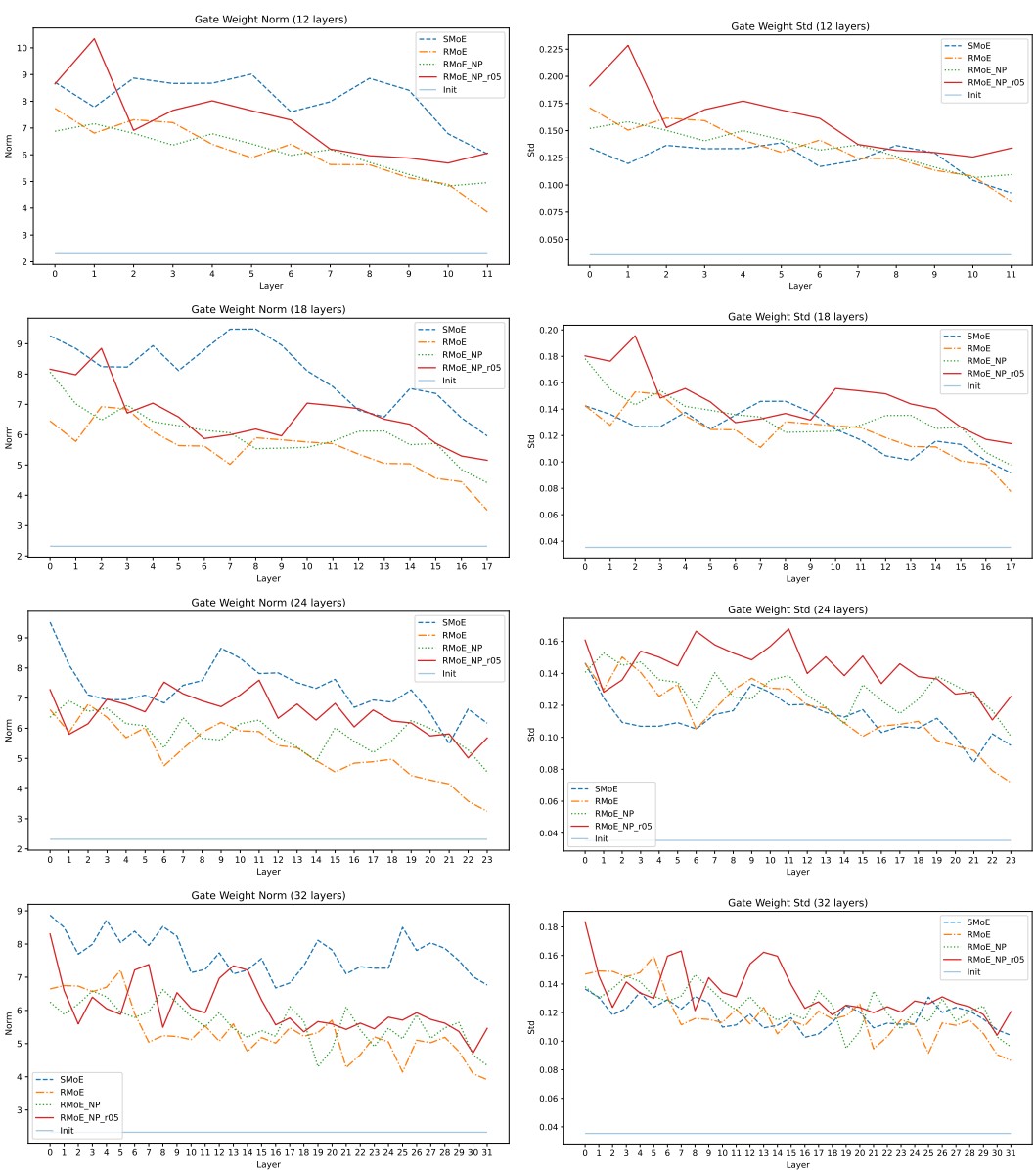

Figure 11: Different layers' router weight statistics (left column: norm and right column: standard deviation) in Enwiki8 setting. (1) different layers have different norms and STDs, which inspires us to introduce layerwise projector in Equ 4 and explains using the shared projector can hurt RMoE's performance (Tab. 6). (2) While SMoE routers show larger weight norms than RMoE settings, their standard deviations are not the highest. The large router norms can potentially explain the larger IB and OB in Tab. 8.

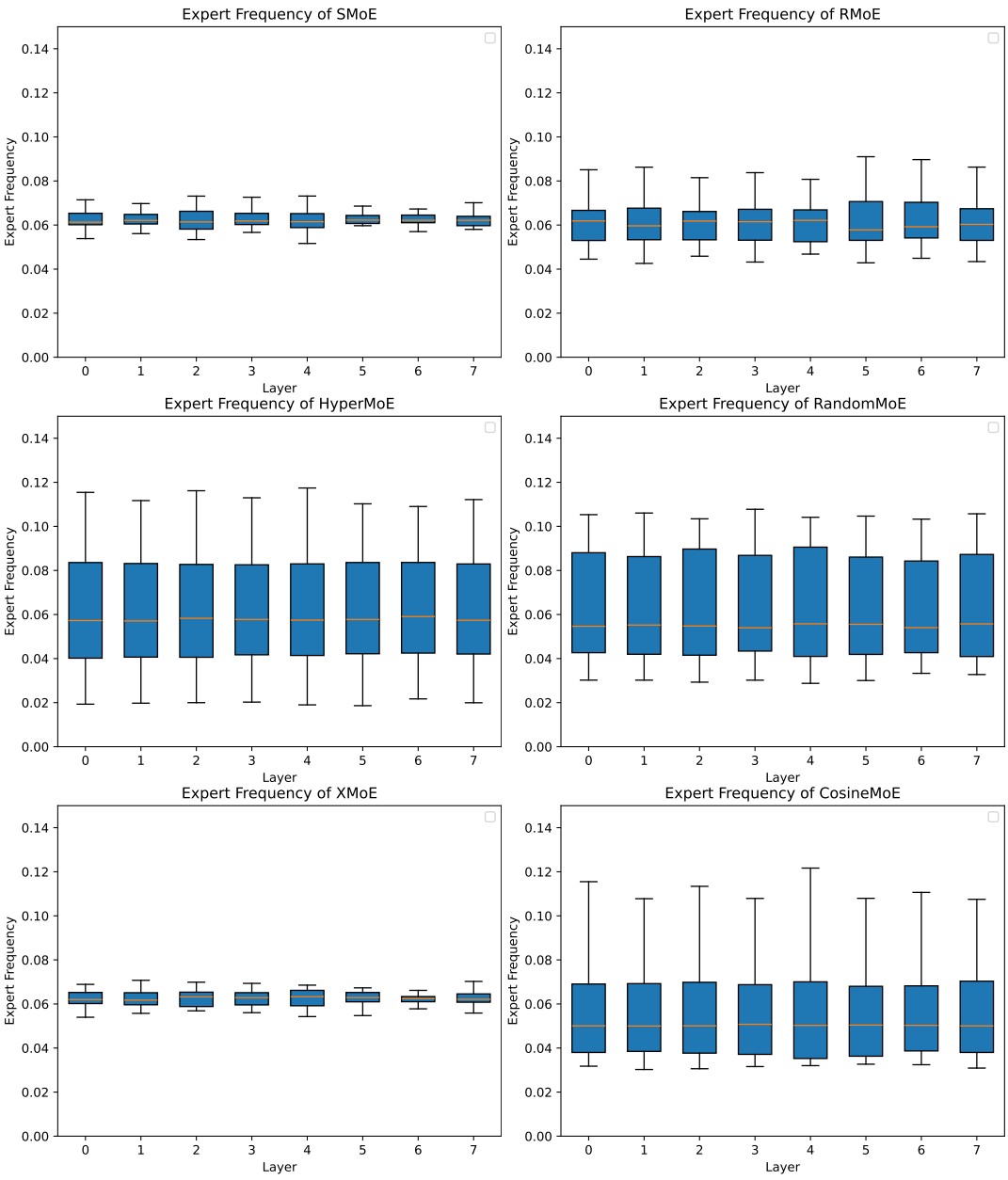

Figure 12: Different methods' expert selection frequency on medium size models in Enwiki8. (1) RMoE slightly increases expert imbalance than SMoE. (2) Methods using a frozen-random-initialize router (Hyper-MoE and RandomMoE) show more imbalance problems.

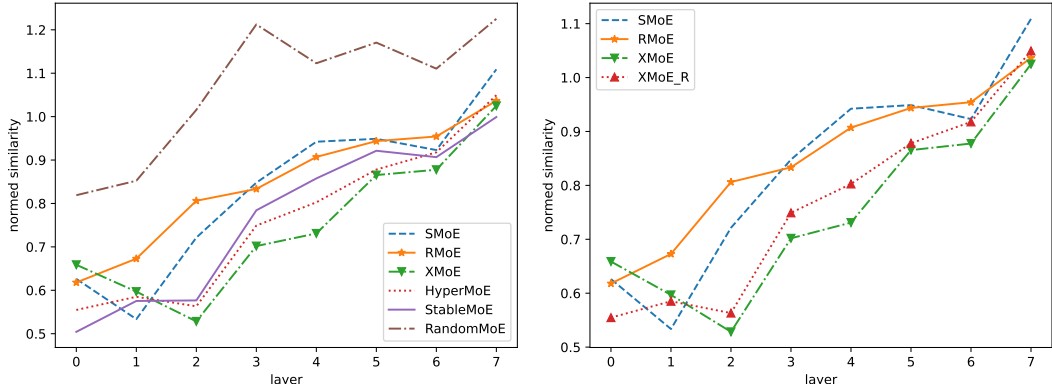

Figure 13: Expert similarity in Enwiki8 training experiments. RandomMoE shows the highest expert similarity. XMoE, which introduces down-projected cosine routing to resolve representation collapse in SMoE, shows the lowest expert similarity. While RMoE doesn't significantly diversify experts as in the large-scale training settings (left), it can be further combined with XMoE, which largely increases expert diversity and brings improvement (right).

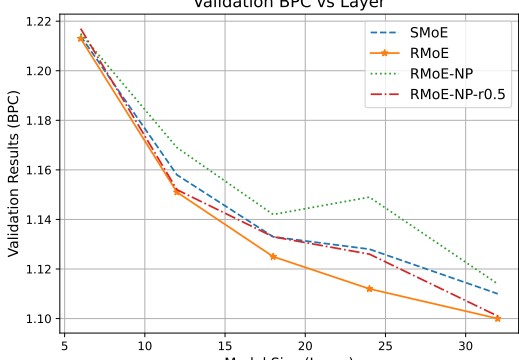

Figure 14: Validation BPC on Enwiki8 with different model sizes (6, 12, 18, 24, 32 layers).

