# OpenReview forum: "Layerwise Recurrent Router for  Mixture-of-Experts"
_ICLR.cc/2025/Conference — ICLR 2025 Poster_

### Official Review · Reviewer_oYLa · 2024-10-29

**Soundness:** 2
**Presentation:** 3
**Contribution:** 3
**Rating:** 8
**Confidence:** 5

**Summary:**

This paper proposes to use RNN to replace the linear projection layer to route the tokens in MoE models. The proposed model achieves better results than vanilla router across scales (after revision).

**Strengths:**

1) Elegant design. Using previous routing representation to improve the routing decision is a smart idea.
2) Some experimental results in Section 6 (Observation) are interesting, such as Layer-wise recurrence encourages expert diversity.
3) Comprehensive ablation studies in Section, which shows the effectiveness of RNN router under this scale.
4) Unshare part of trainable parameters in RNN also makes sense. I suggest to cite this work (https://arxiv.org/abs/2107.11817), which unshare the layernorm when sharing MoE Transformer blocks.

**Weaknesses:**

I have only one important concern, i.e. the scale of the experiments. I agree the paper is promising in many aspects, but the experiments are too toy for a model architecture paper in 2024. We need at least 3B/7B activated parameters and 1T tokens to justify the scalability of the proposed method.

However, I understand it might be difficult to acquire enough computation resources for some reasons, so I decided to give a positive rating to this paper.

**Questions:**

NA

---

> ### Author Response · Authors · 2024-11-21
> **Responses to reviewer oYLa**
>
> Thanks for your review.
>
> To further validate the effectiveness of our method, we supplement experiments with larger scales, i.e., larger models and more training tokens. We **train models up to 43B total parameters with 6.9B activated parameters and 400B tokens**. We evaluate our trained model on MMLU, GSM8K, Hellaswag, and Humaneval. We observe **consistent performance gains** on these tasks compared with the standard MoE (MMLU, 65.8 v.s. 64.4, GSM8K 67.4 v.s. 65.1, human eval 37.8 v.s. 35.4). Although we do not scale tokens to 1T level, the model trained on 400B high-quality data has already achieved comparatively high performance. We humbly believe that experiments with this scale could validate the effectiveness of our method. For more detailed results, please refer to the general comment, common concern 1.
>
> Thanks for the mentioned related work. We have cited it in our revised paper.
>
> Please let us know if there is anything we could do to let you recommend our paper further!

---

> ### Author Response · Authors · 2024-11-23
> **Request for discussion**
>
> Dear reviewer oYLa,
>
> We have posted some experiment results, e.g., more larger scale experiments (from 15B to 43B models trained with 400B tokens) with more significant results (1.35 performance gains on MMLU, 2.27 on GSM8K, and 2.43 on Humaneval). We have also updated our paper according to your corcerns. We would appreciate it if you could let us know whether our responses properly address your concerns.
>
> Look forward to your reply.
>
> Best regards,
>
> Authors

---

> > ### Comment · Reviewer_oYLa · 2024-11-27
> >
> > Thanks for providing more experiments. Looks nice. Not a huge improvement but it is good to see consistent improvement across scales via such a simple modification. I raised my score to 8.

---

> > > ### Author Response · Authors · 2024-11-28
> > >
> > > Thanks for your reply and your recommendation for our paper!

---

### Official Review · Reviewer_Cw7i · 2024-11-03

**Soundness:** 3
**Presentation:** 3
**Contribution:** 2
**Rating:** 6
**Confidence:** 3

**Summary:**

The paper presents the Layerwise Recurrent Router for Mixture-of-Experts (RMoE), designed to address inefficiencies where routers at each layer operate independently, leading to suboptimal token-expert assignments. RMoE integrates a GRU to enable cross-layer communication, establishing dependencies between routing decisions across layers. The authors validate RMoE through experiments across various model sizes and tasks, demonstrating consistent improvements over baseline MoE models. Analysis shows that RMoE’s gains are primarily due to effective cross-layer information sharing, resulting in better expert utilization and diversity.

**Strengths:**

- The use of a GRU to incorporate layerwise information into MoE is a novel and well-motivated approach to address parameter inefficiencies in SMoE.
- Extensive ablation studies demonstrate the importance of recurrent gradient flow, validating the primary contributor to RMoE’s improvements.
- Interesting analysis on the increase in cross-layer mutual information and gating score entropy with RMoE, providing insights into how cross-layer information sharing improves expert selection and model robustness.

**Weaknesses:**

- Current performance improvement appears marginal; val and test losses alone do not demonstrate statistically significant gains over baselines. Benchmarking on standard language modeling tasks would provide more practical context for the model's effectiveness.
- Analysis of gradient flow could be more rigorous; while recurrent gradient flow is highlighted as crucial in Section 5, more sophisticated gradient tracking techniques would strengthen this claim.
- The additional computation stage introduced by the GRU increases complexity, yet the impact on training efficiency is not thoroughly analyzed across different model sizes.

**Questions:**

- Clarify the architecture of the layer-specific projectors used in Eq. 4; are there structural variations across layers, or have alternative projector architectures been tested for potential performance improvements?
- In Table 1, improvement over baselines is minor, yet gains are more noticeable in Table 2 (billion-level settings). Could the authors explain the differing results, specifically regarding training sizes and conditions?
- Column names in Table 8 appear inconsistent with descriptions in the text; could these be corrected or clarified?

---

> ### Author Response · Authors · 2024-11-21
> **Responses to reviewer Cw7i**
>
> > Current performance improvement appears marginal; val and test losses alone do not demonstrate statistically significant gains over baselines. Benchmarking on standard language modeling tasks would provide more practical context for the model's effectiveness.
>
> To further validate the effectiveness of our method, we supplement experiments with larger scales, i.e., larger models and more training tokens. We train models up to **43.1B total parameters and 400B tokens**. We evaluate our trained model on MMLU, GSM8K, Hellaswag, and Humaneval. We observe **significant performance gains on these tasks** compared with the baseline (MMLU, 65.8 v.s. 64.4, GSM8K 67.4 v.s. 65.1, human eval 37.8 v.s. 35.4). We humbly believe that experiments with this scale could validate the effectiveness of our method. For more detailed results, please refer to the general comment, common concern 1.
>
> >  Analysis of gradient flow could be more rigorous; while recurrent gradient flow is highlighted as crucial in Section 5, more sophisticated gradient tracking techniques would strengthen this claim.
>
> Thanks for your suggestion! We track the gradient norm of different routers at different training steps. We empirically found that the introduced recurrent gradient flow could partially prevent the router from being dominated by the load balancing loss through training, thus bringing improvements. We think this extra analysis could strengthen our claim. Please refer to the general comment, Common Concern 2, for more detailed results.
>
> > The additional computation stage introduced by the GRU increases complexity, yet the impact on training efficiency is not thoroughly analyzed across different model sizes.
>
> We reported the training time and memory usage in Table 1 and Table 2 in our submission. The training speed and the memory usage are approximately the same as the SMoE baseline, because the bottleneck remains the calculation of other parameters in the model.
>
> > Clarify the architecture of the layer-specific projectors used in Eq. 4; are there structural variations across layers, or have alternative projector architectures been tested for potential performance improvements?
>
> Thanks for your question! We have improved the clarity of this part in our revised paper. To be clear, each projector is just one single linear layer. By mentioning layer-specific, we are emphasising that the parameters of projectors are not shared across layers.
>
> We empirically validate that employing layer-specific/unshared projectors is beneficial. This may be because the distributions and norms of hidden states in different layers shift. We have not introduced more sophisticated projectors, which we believe is promising but is not the main focus of our paper.
>
> >  In Table 1, improvement over baselines is minor, yet gains are more noticeable in Table 2 (billion-level settings). Could the authors explain the differing results, specifically regarding training sizes and conditions?
>
> The difference between Table 1 and Table 2 is mainly because of metrics. Table 1 uses BPC and PPL, whereas Table 2 uses accuracy. A similar trend is also observed in our supplemented experiments in the large-scale setting, presented in the general comment, common concern 1. The difference in the ppl is usually small, but the improvement for downstream tasks is usually more significant.
>
> > Column names in Table 8 appear inconsistent with descriptions in the text; could these be corrected or clarified?
>
> Thanks for your question! From your description, we guess that the inconsistency may be: in the text description, we formally define the **Inner Balance** with the ratio top1 / top-k; and in the caption of Fig. 8, we mentioned the top1 / top2. This is because we set k=2 in all our experiments. We have revised this part to make it more clear!

---

> ### Author Response · Authors · 2024-11-23
> **Request for discussion**
>
> Dear reviewer Cw7i,
>
> We have posted some experiment results (e.g., more larger scale experiments with more significant results and direct analysis of router gradient norm) and clarifications to answer your questions. We have also updated our paper according to your suggestions. We would appreciate it if you could let us know whether our responses properly address your concerns.
>
> Look forward to your reply.
>
> Best regards,
>
> Authors

---

> > ### Comment · Reviewer_Cw7i · 2024-11-26
> >
> > The authors have addressed most of my concerns. I will update my score accordingly.

---

> > > ### Author Response · Authors · 2024-11-28
> > >
> > > Thanks for your reply and your recommendation for our paper!

---

### Official Review · Reviewer_bumc · 2024-11-06

**Soundness:** 2
**Presentation:** 2
**Contribution:** 2
**Rating:** 3
**Confidence:** 4

**Summary:**

Paper introduces a gated recurrent connection for routing decisions in MoEs such that routing decision in a layer depend on shared GRU so that routing decisions between layers are no longer independent. The intuition expressed by authors is that this should help improve model because routing decisions in current layer should be able to leverage routing info from previous layers.

**Strengths:**

Paper clearly expresses main idea, that routing decisions in MoE should have some dependency on previous routing decisions, and how they achieve this and test the results.

**Weaknesses:**

The results are relatively minor improvements (1-2% relative), the models are very small (on the order of 36M parameters for an MoE model) and the benchmarks are small (enwiki8 & wikitext).

**Questions:**

Suggestions for improvement are to show results on a larger MoE model. Not all models need to be 100s of billions of parameters, but a 36M parameter MoE model is quite small these days, and it is hard to be convinced that small gains on this model would translate to a more realistic sized model.

---

> ### Author Response · Authors · 2024-11-21
> **Responses to Reviewer bumc**
>
> >  Larger MoE models
>
> Thanks for your suggestion. In our submission, we showed the results not only on the 36M small model, but also the results with approx. 1B model trained with 20B / 40B tokens. To further validate the effectiveness of our method, we supplement experiments with larger scales, i.e., larger models and more training tokens. We train models up to **43.1B total parameters and 400B tokens**. We evaluate our trained model on MMLU, GSM8K, Hellaswag, and Humaneval. We observe **significant performance gain on these tasks compared with the baseline** (MMLU, 65.8 v.s. 64.4, GSM8K 67.4 v.s. 65.1, human eval 37.8 v.s. 35.4). We humbly believe that experiments with this scale could validate the effectiveness of our method. For more detailed results, please refer to the general comment, common concern 1.

---

> ### Author Response · Authors · 2024-11-23
> **Request for discussion**
>
> Dear reviewer bumc,
>
> We have posted some experiment results, e.g., more larger scale experiments (from 15B to 43B models trained with 400B tokens) with more significant results (1.35 performance gains on MMLU, 2.27 on GSM8K, and 2.43 on Humaneval). We have also updated our paper according to your corcerns. We would appreciate it if you could let us know whether our responses properly address your concerns.
>
> Look forward to your reply.
>
> Best regards,
>
> Authors

---

> ### Author Response · Authors · 2024-11-28
> **Reminder for discussion**
>
> Dear reviewer bumc,
>
> We apologize for reaching out with another reminder of the discussion.
>
> We want to highlight that we have provided additional experimental results following your suggestions.
>
> Specifically, we pre-trained models with up to **43.1B total parameters** for up to **400B tokens** and evaluated the models on MMLU, gsm8k, HumanEval, etc.
>
> We observe significant improvements in these tasks.
>
> We would appreciate your thoughts on whether these updates satisfactorily address your concerns.
>
> Best,
>
> Authors

---

> ### Author Response · Authors · 2024-12-02
>
> Dear Reviewer bumc,
>
> We apologize for reaching out with another reminder as the discussion period is coming to an end.
>
> We have updated our manuscript with results from larger-scale experiments. We are pleased to note that Reviewer oYLa, who initially raised concerns about the experimental scale, has indicated that these concerns have been addressed and has raised their score from 6 to 8.
>
> We hope that these new results will help address your concerns regarding the experiments and look forward to further discussions with you.
>
> Thank you for your attention.
>
> Best regards

---

### Official Review · Reviewer_rPzv · 2024-11-12

**Soundness:** 3
**Presentation:** 3
**Contribution:** 3
**Rating:** 6
**Confidence:** 4

**Summary:**

This paper proposes the Layerwise Recurrent Router for Mixture-of-Experts, a novel approach to enhance efficiency in MoE architectures within LLM. The authors highlight limitations in traditional MoE routers that operate independently across layers, potentially resulting in suboptimal parameter utilization. To address this, Recurrent-Router MoE introduces a GRU-based mechanism that enables dependencies between routing decisions across layers, which improves the diversity and effectiveness of expert utilization. The proposed method integrates seamlessly with other MoE architectures and demonstrates substantial improvements in performance on various language modeling tasks. Through comprehensive experiments, the authors show that RR-MoE achieves superior performance over several baseline router designs with only marginal computational overhead.

**Strengths:**

The paper offers a simple but new enhancement to MoE architectures by introducing a layerwise recurrent mechanism, distinguishing it from other MoE router designs that operate independently across layers. The methodology is sound, with a well-motivated design backed by extensive experimental validation and ablation studies. The work is valuable to the research community, as it not only provides a means to improve existing MoE designs but also contributes insights into the impact of cross-layer recurrence on expert diversity and selection.

**Weaknesses:**

The authors mention that their model is under-trained and does not yet perform well on challenging tasks like MMLU and GSM8K. A more complete evaluation on these benchmarks, perhaps by including more training tokens or longer training duration, would provide a fuller picture of the model's real-world applicability.

Also the paper presents extensive performance metrics and ablation studies, a more direct analysis of gradient propagation through the recurrent router could strengthen the argument for the importance of recurrent gradients in optimizing routing decisions. Also, do we need truncated backpropagation similar to BPTT?

The paper could benefit from a deeper comparative analysis with recent MoE models that also use recurrence or other advanced routing strategies (such as XMoE, HyperMoE etc). This would help clarify how RRMoE’s benefits compare to similar innovations in this rapidly evolving field.

**Questions:**

The paper emphasizes the importance of gradient flow through the GRU-based recurrence in RMoE but does not mention whether Truncated Backpropagation Through Time (BPTT) or any specific method was used to manage gradients across layers. Did the authors consider using truncated BPTT for controlling gradient flow through the recurrent router? Including details on gradient management would help clarify if additional methods, like truncation, could further optimize training.

 Would it be possible to conduct a more direct analysis, such as by visualizing or measuring gradient norms through layers, to further validate the specific impact of recurrent gradients?

 Could the authors provide insights or quantitative analysis on how the recurrence depth impacts the RMoE’s routing decisions?

The current evaluation omits challenging benchmarks like MMLU and GSM8K, mentioning that the models are still under-trained. Can the authors comment on whether they plan to scale the training or incorporate task-specific fine-tuning to gauge RMoE’s performance on these harder tasks? Additional results on these benchmarks could help better gauge RMoE’s robustness and adaptability in real-world scenarios.

---

> ### Author Response · Authors · 2024-11-21
> **Responses to Reviewer rPzv**
>
> > More complete evaluation.
>
> We have provided the results of training larger models on more tokens (15B models on 120B tokens, 15B models on 400B tokens, and 43B models on 400B tokens). Our proposed RMoE network demonstrates substantial improvements regarding training perplexity and downstream task performance. When training the **43B MoE with 400B tokens, RMoE achieves 1.35 performance gains on MMLU, 2.27 on GSM8K, and 2.43 on Humaneval compared with the standard SMoE**. The experimental details are explained as presented in the General Comment Common Concern 1.
>
> >  A more direct analysis
>
> Please refer to our general comment, common concern 2. We analyse the grad norm and the drop_ratio of different routers through the training. We validate that the extra recurrent gradient effectively reduces the influence of the load balancing loss on router training, therefore bringing improvements.
>
> > The paper could benefit from a deeper comparative analysis with recent MoE models that also use recurrence or other advanced routing strategies (such as XMoE, HyperMoE etc). This would help clarify how RRMoE’s benefits compare to similar innovations in this rapidly evolving field.
>
> Thanks for your suggestion. To the best of our knowledge, we are the first to introduce cross-layer recurrence into the MoE architecture and provide an analysis of gradient flow. We respectfully believe that the ablations in our submission and additional experimental results in our rebuttal message could well explain how RMoE benefits: an extra gradient pathway!
>
> > The paper emphasizes the importance of gradient flow through the GRU-based recurrence in RMoE but does not mention whether Truncated Backpropagation Through Time (BPTT) or any specific method was used to manage gradients across layers. Did the authors consider using truncated BPTT for controlling gradient flow through the recurrent router? Including details on gradient management would help clarify if additional methods, like truncation, could further optimize training.
>
> Thanks for your question. We have not used *truncated* BPTT or any other specific method to manage the gradients across layers. The GRU is introduced to capture the dependency of routing decisions between different layers, and the largest model we train (43.1 B parameters) has 32 layers, which is still a relatively short sequence for a standard GRU model.
>
> >  Would it be possible to conduct a more direct analysis, such as by visualizing or measuring gradient norms through layers, to further validate the specific impact of recurrent gradients?
>
> Thanks for your suggestion. We have provided a more direct analysis of the recurrent gradient's impact. We investigate how routers' gradient norms and drop ratio vary during training. We validated that the extra recurrent gradient path could help prevent the router from overfitting the load balancing loss during the early training stage. Please refer to our general comment, common concern 2, for more detailed results.
>
> >  Could the authors provide insights or quantitative analysis on how the recurrence depth impacts the RMoE’s routing decisions?
>
> We humbly believe that the recurrence depth does not have much impact on GRU to capture dependencies of routing decisions across layers. This is because the models are usually not that deep. Even our trained 43.1B models only have 32 layers, which is still a relatively shorter sequence for the GRU model. Moreover, we think the recurrence depth mainly helps the gradient backward propagation. In section 5, we find that only using recurrency in the forward process will not improve the performance.
>
> >  The current evaluation omits challenging benchmarks like MMLU and GSM8K, mentioning that the models are still under-trained. Can the authors comment on whether they plan to scale the training or incorporate task-specific fine-tuning to gauge RMoE’s performance on these harder tasks? Additional results on these benchmarks could help better gauge RMoE’s robustness and adaptability in real-world scenarios.
>
> Thanks for your questions. To provide further empirical results, we train models up to **43.1B total parameters and 400B tokens**. We evaluate our trained model on MMLU, GSM8K, Hellaswag, and Humaneval. We observe significant performance gains on these tasks compared with the baseline (MMLU, 65.8 v.s. 64.4, GSM8K 67.4 v.s. 65.1, human eval 37.8 v.s. 35.4). We humbly believe that experiments with this scale could validate the effectiveness of our method. For more detailed results, please refer to the general comment, common concern 1.

---

> ### Author Response · Authors · 2024-11-23
> **Request for discussion**
>
> Dear reviewer rPzv,
>
> We have posted some experiment results (e.g., more larger scale experiments with more significant results and direct analysis of router gradient norm) and clarifications to answer your questions. We have also updated our paper according to your suggestions. We would appreciate it if you could let us know whether our responses properly address your concerns.
>
> Look forward to your reply.
>
> Best regards,
>
> Authors

---

> ### Author Response · Authors · 2024-11-28
> **Reminder for discussion**
>
> Dear reviewer rPzv,
>
> We apologize for reaching out with another reminder of the discussion.
>
> We want to highlight that we have made significant revisions to our paper following your suggestions, including more complete, large-scale evaluations, a more direct analysis of the recurrent gradient, and other clarification responses to your questions.
>
> We wonder if it is possible for you to let us know if our responses could address your concerns. We would appreciate your thoughts on whether these updates satisfactorily address the theme.
>
> Best,
>
> Authors

---

> ### Author Response · Authors · 2024-12-02
>
> Dear Reviewer rPzv,
>
> We apologize for reaching out with another reminder as the discussion period is coming to a close.
>
> We have updated our manuscript to include more direct analysis of the router gradients. We are pleased to note that Reviewer Cw7i, who initially raised concerns about this analysis, has indicated that these concerns have been addressed and has raised their score.
>
> We hope that these new results will help address your concerns, and we look forward to further discussions with you.
>
> Thank you for your attention.
>
> Best regards

---

### Author Response · Authors · 2024-11-21
**General Comment Part 1 / 4**

We sincerely appreciate all the reviewers' time and effort in reviewing our paper. We are glad to find that reviewers generally recognise our strengths/contributions:

1. We introduce the cross-layer recurrence into MoE networks to capture the dependency of routing decisions in different layers, which is novel and well-motivated (**rPzv, Cw7i, oYLa**).
2. We provide extensive ablation studies to identify the primary factors contributing to RMoE’s performance improvements, presenting interesting and insightful experimental results (**rPzv, Cw7i, oYLa**).

We thank the reviewers for their suggestions for improving our paper. In addition to point-by-point responses for each reviewer, we address the common concerns raised by the reviewers as follows.

>  Common concern 1: Experiments scale. Reviewers (**rPzv, bumc, oYLa**) are concerned about our proposed method's effectiveness when applied to a larger scale, and reviewer Cw7i encourages us to provide more practical context for the model’s effectiveness.

To summarise, we extend our experiments for training MoE models up to **43.1B total parameters with 400B tokens**. Our proposed RMoE network demonstrates substantial improvements in perplexity and downstream task performance. Compared with the standard SMoE, RMoE achieves **1.35 performance gains on MMLU, 2.27 on GSM8K, and 2.43 on Humaneval**.

Specifically, we ran extra experiments with the following settings:

1. Models with 15B total parameters and 2.54B activated parameters, trained with 120B tokens.
2. Models with 15B total parameters and 2.54B activated parameters, trained with 400B tokens.
3. Models with 43.1B total parameters and 6.87B activated parameters, trained with 400B tokens.

In settings 1 and 2, we evaluate different router variants (baseline linear router, MLP router, and our proposed RNN router) on the training perplexity, hellaswag, and challenge tasks like MMLU, and GSM8K. In the setting 3, which requires far more computational resources, we compare the linear router and the RNN router in terms of training perplexity, hella swag, MMLU, GSM8K, and humaneval. More details about the experiment are updated in the appendix part (Appendix A.3 to A.5).

The gist of our supplemented experiment setting is as follows:

1. We compare the different router variants in each setting. We also provide the intermediate results at different training steps to further demonstrate the robustness of improvement.
2. With settings 1 and 2, we compare results with different training tokens (120B vs. 400B) to validate the scaling ability with data.
3. With the setting 2 and the setting 3, we compare results with different model scales. (15B v.s. 43.1B) to justify the scaling ability with model size.

Our results are summarised in the following tables:

| Setting: 15B activate 2.54B,  120B tokens | Hellaswag | MMLU      | GSM8K     | training ppl |
| ----------------------------------------- | --------- | --------- | --------- | ------------ |
| 40k steps (80B tokens)                    |           |           |           |              |
| linear router                             | 67.69     | 46.24     | 24.18     | 7.406        |
| MLP router                                | 67.98     | 46.47     | 23.58     | 7.437        |
| RNN router (Ours)                         | **68.00** | **47.74** | **27.14** | **7.361**    |
| 50k steps (100B tokens)                   |           |           |           |              |
| linear router                             | 70.98     | 50.61     | 30.78     | 6.754        |
| MLP router                                | 70.80     | 50.6      | 30.17     | 6.786        |
| RNN router (Ours)                         | **71.02** | **51.74** | **32.98** | **7.361**    |
| 60k steps (120B tokens)                   |           |           |           |              |
| linear router                             | 72.03     | 52.79     | 34.80     | 6.447        |
| MLP router                                | 72.19     | 52.81     | 34.57     | 6.479        |
| RNN router (Ours)                         | **72.36** | **54.02** | **36.13** | **6.425**    |

---

> ### Author Response · Authors · 2024-11-21
> **General Comment Part 2 / 4**
>
> | Setting: **15B activate 2.54B param. Training 400B tokens** | Hellaswag | MMLU      | GSM8K     | training ppl |
> | ----------------------------------------------------------- | --------- | --------- | --------- | ------------ |
> | 50k steps (200B tokens)                                     |           |           |           |              |
> | linear router                                               | 69.48     | 49.96     | 33.21     | 7.718        |
> | MLP router                                                  | 69.76     | 50.27     | 31.77     | 7.736        |
> | RNN router (Ours)                                           | **70**    | **52.21** | **32.98** | **7.608**    |
> | 70k steps (280B tokens)                                     |           |           |           |              |
> | linear router                                               | 72.4      | 54.66     | 42.61     | 6.477        |
> | MLP router                                                  | 72.62     | 55.33     | 38.51     | 6.502        |
> | RNN router (Ours)                                           | **73.18** | **56.06** | **44.35** | **6.4**      |
> | 100k steps (400B tokens)                                    |           |           |           |              |
> | linear router                                               | 76.39     | 59.54     | 52.16     | 5.685        |
> | MLP router                                                  | 76.09     | 59.96     | 51.71     | 5.709        |
> | RNN router (Ours)                                           | **76.72** | **60.6**  | **52.99** | **5.62**     |
>
> | Setting: **43.1B activate 6.9B param. Training 400B tokens** | Hellaswag | MMLU      | GSM8K     | Humaneval | training ppl |
> | ------------------------------------------------------------ | --------- | --------- | --------- | --------- | ------------ |
> | 50k steps (200B tokens)                                      |           |           |           |           |              |
> | linear router                                                | 74.47     | 56.1      | 47.01     | 28.05     | 6.184        |
> | RNN router (Ours)                                            | **74.96** | **57.21** | **48.6**  | **30.49** | **6.104**    |
> | 70k steps (280B tokens)                                      |           |           |           |           |              |
> | linear router                                                | 77.13     | 61.37     | 56.1      | 28.66     | 5.585        |
> | RNN router (Ours)                                            | **77.95** | **62.42** | **56.86** | **31.71** | **5.521**    |
> | 90k steps (360B tokens)                                      |           |           |           |           |              |
> | linear router                                                | 79.2      | 64.34     | 64.97     | 34.15     | 5.101        |
> | RNN router (Ours)                                            | **79.71** | **65.45** | **65.96** | **37.2**  | **5.047**    |
> | 100k steps (400B tokens)                                     |           |           |           |           |              |
> | linear router                                                | 79.62     | 64.41     | 65.13     | 35.37     | 4.995        |
> | RNN router (Ours)                                            | **80.29** | **65.76** | **67.4**  | **37.80** | **4.936**    |
>
> The tables validate that **our proposed method is still effective when training larger MoE language models with more tokens**. The empirical findings are consistent with those of our small-scale settings.
>
> 1. Our proposed RNN router outperforms the linear and MLP routers in all settings, training steps, and tasks, demonstrating significant improvements in practical scenarios.
> 2. Although the MLP router adds more parameters to the original linear router, it still underperforms the simple linear router on most tasks. This indicates that simply involving more computation in the router stage cannot boost the MoE’s performance. Our results in Table 1 of the paper support this observation.

---

> > ### Author Response · Authors · 2024-11-21
> > **General Comment 3 / 4**
> >
> > > Common concern 2: Further gradient analysis. We provided the ablation results to highlight the importance of the router gradient in our submission. The reviewer rPzv suggests conducting a more direct analysis. The reviewer Cw7i indicates that the analysis of the gradient flow could be more rigorous.
> >
> > We have supplemented extra analysis about the gradient norm of our proposed routers and the linear routers.
> >
> > Specifically, based on the setting of training 15B models for 120B tokens, we investigate how the gradient norm of the router varies throughout the entire training process. When training an MoE-based model, the gradient of the router has two separate sources: due to (1) the language modeling loss, and (2) the load balancing (lb) loss that forces the router to assign tokens to different experts in a balanced manner. Therefore, for each router, we compare the gradient from the language modelling loss only and from the whole training loss. We calculate the average for 100 training steps to estimate the gradient norm.
> >
> > Furthermore, to better investigate the relation between the router behavior and the router gradient, we calculate the **drop ratio** for the router. This is because during the large-scale MoE pre-training, to ensure the training efficiency, the expert is usually controlled by a hyper-parameter called *capacity factor*, which determines the total tokens that one expert can process. If the router assigns tokens to some expert that exceeds its capacity, the expert will drop tokens with the lowest scores. And we define the **drop_ratio** as tokens dropped / total tokens. The lb loss mentioned before is critical to decreasing the drop_ratio.
> >
> > The results are summarised in the following table. There are eight lines:
> >
> > 1. The gradient of the linear router from the whole training loss;
> > 2. The gradient of the linear router from the language modeling loss;
> > 3. The gradient of the linear router from the lb loss: line1 - line2
> > 4. The drop_ratio of the linear router;
> > 5. The gradient of the RNN router from the whole training loss;
> > 6. The gradient of the RNN router from the language modeling loss;
> > 7. The gradient of the RNN router from the lb loss: line 5 - line 6;
> > 8. The drop_ratio of the RNN router.
> >
> > | Training steps (k step, 20B tokens) | 0.1    | 10     | 20     | 30     | 40    | 50    | 60    |
> > | ----------------------------------- | ------ | ------ | ------ | ------ | ----- | ----- | ----- |
> > | Linear router                       |        |        |        |        |       |       |       |
> > | grad from the whole loss            | 1.058  | 0.1938 | 0.1911 | 0.198  | 0.208 | 0.217 | 0.221 |
> > | grad from LM loss                   | 0.6252 | 0.1833 | 0.1836 | 0.1917 | 0.204 | 0.215 | 0.220 |
> > | grad from lb loss                   | 0.4328 | 0.0105 | 0.0075 | 0.0063 | 0.004 | 0.002 | 0.001 |
> > | drop ratio                          | 35.6   | 5.43   | 5.34   | 5.17   | 4.89  | 4.64  | 4.5   |
> > | RNN router (Ours)                   |        |        |        |        |       |       |       |
> > | grad from the whole loss            | 0.9722 | 0.1602 | 0.1529 | 0.1534 | 0.155 | 0.155 | 0.154 |
> > | grad from LM loss                   | 0.6358 | 0.146  | 0.1379 | 0.1392 | 0.144 | 0.148 | 0.151 |
> > | grad from lb loss                   | 0.3372 | 0.0142 | 0.015  | 0.0142 | 0.011 | 0.007 | 0.003 |
> > | drop ratio                          | 38.7   | 6.35   | 6.3    | 5.94   | 5.32  | 4.54  | 4.09  |
> >
> > According to the table, we have the following observations:
> >
> > 1. The drop_ratio decreases with training for both routers, indicating that they learn to attribute tokens in a balanced manner.
> > 2. According to the drop ratios, we observe the significant behavioural difference between the two routers: during the early training phase (10k steps -> 30k steps), the drop_ratio of the linear router is noticeably lower than that of the RNN router; the drop_ratio of the RNN router archives at the lower value in the end.
> > 3. The trend observed in the drop ratio is consistent with the results of the gradient norm. The gradient norm for load-balancing loss is relatively higher in the RNN router until the final training stage (50k—60k), whereas the gradient from load-balancing loss in the linear router is high at the beginning and generally low from then on (10k—60k).
> >
> > These phenomena indicate that the LB loss could dominate the training of the linear router: when the drop ratio is low and stays unchanged, the grad from LB loss will be low because the router is already well-optimized for LB loss. Such early convergence in the LB loss may reach a suboptimal solution in the trade-off between optimising load balance and language modelling. On the contrary, the gradient of the RNN router from LB loss stabilises in the early training steps (10k - 30k), and the gradient from the lm loss keeps decreasing, suggesting that the RNN router is more optimised towards the LM loss.

---

> > > ### Author Response · Authors · 2024-11-21
> > > **General Comment 4 / 4**
> > >
> > > Overall, the supplemented analysis about the grad norm and drop_ratio suggests that the introduced recurrent router effectively controls the influence of the lb loss. Our findings are also supported by the related works, which reveal that the router optimised more for lb loss can effectively decrease the drop_ratio but at the cost of overall model performance [1].
> > >
> > > We thank the reviewers for their insightful reviews. We have revised our paper to include these extra results (appendix A.3 and A.4.1, marked as blue). We hope this clarifies your concerns and helps you assess our paper.
> > >
> > > [1] Wang, Lean, et al. "Auxiliary-Loss-Free Load Balancing Strategy for Mixture-of-Experts." *arXiv preprint arXiv:2408.15664* (2024).

---

### Meta-Review · Area_Chair_2YTy · 2024-12-22

**Metareview:**

The authors propose changing the standard MoE architecture to allow for direct dependencies between routing decisions across layers. It is a simple idea that is well-executed, giving rise to modest but consistent improvements across a variety of tasks and model scales. Reviewers initially questioned the authors on only running experiments on small model scales as well as on some ablations, but the authors responded with additional experiments (in particular on larger models) that convinced most of the reviewers. Overall, I think this is a useful addition to the growing MoE literature and will be helpful guidance for practitioners seeking to train their own MoEs, and I am happy to recommend acceptance.

I encourage the authors to take the reviewer feedback into account and in particular to move the experiments on larger models (currently in the appendix) to the main text.

**Additional Comments On Reviewer Discussion:**

Reviewers initially questioned the authors on only running experiments on small model scales as well as on some ablations, but the authors responded with additional experiments (in particular on larger models) that convinced most of the reviewers. Reviewer bumc did not reply to the authors but I believe their objections were adequately responded to.

---

### Decision · Program_Chairs · 2025-01-22

Accept (Poster)